# Behavior-Aware Off-Policy Selection in High-Stake Human-Centric Environments

## Abstract

In many human-centric environments, such as education and healthcare, the unobservability of human underlying states has been recognized as a key obstacle for understanding individual needs, thus hindering out ability to provide personalized decision-making policies. Several reinforcement learning (RL)-related approaches have been used to facilitate sequential decision-making in these settings, including off-policy selection (OPS), which aids in safely evaluating and selecting optimal policies offline. However, existing OPS algorithms are unsuitable when both the state is unobserved and the setting requires a personalized policy. To address this challenge, we propose a behavior-aware adaptive policy selection framework (HBO) that first captures potentially unique characteristics of the state from human behaviors, and then estimates when and how to intervene with less uncertainty in a timely manner, with bounded error. HBO is evaluated over two real-world human-centric applications, intelligent tutoring and sepsis treatments, where it significantly enhanced participants' long-term course outcomes and survival rates. Broadly, our work enables improved policy personalization in high-stakes domains where extensive evaluation is not possible.

## 1 Introduction

There is significant interest in using reinforcement learning (RL) in human-centric systems (HCSs), such as healthcare and education, to improve downstream outcomes (Namkoong et al., 2020; Gao et al., 2020; Chi et al., 2011; VanLehn, 2006; Abdelshiheed et al., 2023; Mandel et al., 2014; Ruan et al., 2024; Gottesman et al., 2019). The ultimate goal is personalized, adaptive policies that support each person's needs, but achieving this faces several challenges in practice. Evaluating RL policies online in HCSs is risky and may require a long time horizon to observe results (*e.g.*, several years for clinical trials). Off-policy evaluation (OPE) and selection (OPS) mitigate this by using historical data to assess policies before deployment (Jiang & Li, 2016; Fu et al., 2021; Thomas & Brunskill, 2016; Yang et al., 2022; Gao et al., 2022). However, collecting HCS data is usually time and resource intensive (Gao et al., 2024a) and data is generally limited, which motivates the need for uncertainty-aware, data-efficient estimators. Moreover, HCS deployments often face external protocols and constraints that restrict which policies can be considered. Only a (small) subset of the candidate policies that are likely to be beneficial to the *overall* trial population are approved to be deployed, making careful off-policy selection necessary, rather than unrestricted policy learning.

Existing OPE/OPS techniques typically assume that the policy is Markov in the observed features. In HCSs, applying the same policy to different participants may not result in the same expected outcome, *e.g.*, patients diagnosed with the same type of cancer may have very different outcomes from the same treatment policy, given the patients' varied medical history and health conditions which may not be observable. While OPE methods (e.g., importance sampling or doubly-robust estimators (Precup, 2000; Jiang & Li, 2016)) can account for rewards and dynamics that depend on the full history of features, off-policy selection and optimization typically focus on Markov decision policies that condition only on the same immediate state, implicitly assuming identical decisions for all individuals with the same current features.

In HCSs, we are interested in supporting the best outcomes for a new individual. Unlike some other settings, in HCSs *each individual is exposed to a given policy only once*– there are no resets to rewind time and treat a patient with a different initial policy or teach a student fractions from

scratch in a new way. But there can be an opportunity to use observations from interacting with a new individual to help inform and select the best policy, *from the set of approved policies overseen by related departments*, to maximize their expected outcomes. That is contrast with pure OPS, which commits to one policy before any interaction.

To address these challenges, we introduce Human-centric Behavior-aware adaptive Off-policy selection (**HBO**). HBO uses historical data and evaluates online observations with a new individual under a pre-approved acceptable policy, to decide *if* and *when* to switch to a new policy to optimize individual outcomes. We notice that if critical behavioral changes are observed about a new participant, that may enable the system to select a much better decision policy that will maximize expected outcomes for that individual, rather than selecting a single decision policy to be used for all individuals. For example, in personalized medicine only a limited set of approved treatments (*e.g.*, chemotherapy drugs) exists for a given disease, and clinicians often adapt a patient's regimen based on their response to maximize outcomes.

Specifically, our approach first discovers critical temporal behavioral patterns (CBPs) in historical trial data that indicate when a policy switch could be beneficial (*e.g.*, early signs of disease progression). For a new individual under a default approved policy, we monitor their real-time behavior to detect any CBPs. When a CBP occurs, our algorithm estimates the expected outcomes of switching the individual to each of a finite set of alternate approved policies (or decides to wait for more data), and operates policy switching based on confidence. To the best of our knowledge, the most related work is first-glance OPS (Gao et al., 2024b) which assigns policies to new participants according to the sub-grouping of their initial observations on the first time step. After the policy is assigned, it is used for that participant for the remainder of the planning horizon. In contrast, our approach is more flexible since it can actively select when and if to switch someone, during their trajectory.

The key contributions of this work are summarized as follows: (**i**) We introduce a novel framework, HBO, that addresses the challenges of partial observability and personalization simultaneously in OPS for HCSs, with the goal of improving outcomes for participants in ongoing trials. Our approach identifies CBPs from historical data, enabling real-time monitoring of participants to assess the potential benefits from revising their treatment plans (*i.e.*, switch to another pre-approved policy), while incorporating a confidence-informed decision mechanism to determine an appropriate timing for switching. (**ii**) We conduct extensive case studies to evaluate HBO in intelligent education and healthcare applications, using data collected from a real-world intelligent tutoring system and MIMIC-III (Johnson et al., 2016) respectively. Our results show that HBO can effectively enhance long-term outcomes of participants compared to baseline approaches. (**iii**) The design and mechanism behind the HBO framework allows it to be readily available to real-world human trials, given the clear motivation and straightforward adaptation of building blocks available.

## 2 RELATED WORK

**OPE/OPS.** OPE and OPS refer to techniques used to estimate the cumulative returns of target policy candidates using historical experience (*i.e.*, offline data) collected from a distinct behavior policy. This can enable easier identification of a subset of (target) policies that can lead to promising returns once deployed. Standard techniques use importance sampling (IS) (Horvitz & Thompson, 1952; Precup et al., 2000), direct method (DM) (Harutyunyan et al., 2016; Li et al., 2010), doubly robust (DR) (Jiang & Li, 2016; Thomas & Brunskill, 2016) and distributional correction estimation (DICE) (Yang et al., 2020; Nachum et al., 2019a; Zhang et al., 2020; Nachum et al., 2019b) approaches. Most existing OPE/OPS techniques are designed toward evaluating homogeneous agents that share largely similar specifications, without any online interactions. Our work leverages both offline dataset and the online observations to select policy for a new individual, aiming to maximize individual outcomes.

**Partial observability in offline RL.** Most RL algorithms assume that the state is fully known and base policy optimization process on this. However, in HCSs, this is unlikely to be true. For example, in patient treatment settings in clinical scenarios, we can only measure values such as vitals and lab values. However, this represents only a portion of the underlying patient state which is largely unobserved. Some works have attempted to account for this partial observability while learning optimal policies including Hidden Parameter MDPs (HiP-MDPs) (Doshi-Velez & Konidaris, 2013; Fu et al., 2023), and partially observed MDPs (POMDP) (Kaelbling et al., 1998), which assume that the underlying parameter representing the agent state is unknown. However, the HiP-MDP and POMDP frameworks in general maximize returns averaged over long horizons. In contrast, in HCSs,

the trials are considered high-stake where immediate evaluations and actions are expected to be taken once needed, *e.g.*, emergency events in ICU. Moreover, most work along the line of HiP-MDP and POMDP are built with the goal of *learning a single optimal policy* that fits a population overall, and rely on sufficient historical experience for capturing environmental dynamics. In contrast, our goal is to adaptively assign pre-trained policies based on information collected from an individual in an ongoing trial. As a result, limited historical data collected from previous cohorts can be used, as well as little prior information given about the individuals in the current cohort, *i.e.*, mostly only the initial observations.

**Personalization in policy learning.** There are a few existing work learning personalized policies by considering individual characteristics and assuming the state is fully known (Hallak et al., 2015; Modi et al., 2018; Sun et al., 2024). For instance, Hallak et al. (2015) propose Contextual MDP and Cluster-Explore-Classify-Exploit framework, of a multi-task setting where tasks are drawn from related MDPs, and the goal is to quickly learn to perform well in each new MDP. However, they generally assume that all exploration policies are acceptable in each new task– for example, Hallak et al. (2015) analyze using random exploration in the "Explore" step, and Modi et al. (2018) use a variant of tabular R-max when executing in a new task. However, in many HCSs, the underlying human state can be largely unobserved, and we often have important restrictions on the policies that can be deployed with each new student or patient, to ensure safety and performance. It is not acceptable to run any possible exploration policy, and pure exploration or the optimization under uncertainty policies (which may take risky actions due to their potential high performance) used by Hallak et al. (2015); Modi et al. (2018); Sun et al. (2024) are unlikely to be allowed by stakeholders.

## 3 HUMAN-CENTRIC BEHAVIOR-AWARE ADAPTIVE OFF-POLICY SELECTION

Due to the high-stakes nature of HCSs, policies used to interact with new participants typically require careful review and approval by experts *a priori* (Gao et al., 2024b). Moreover, anchoring on a single policy over the entire horizon may sacrifice the optimality of the final outcome, as the policy cannot be personalized to participant characteristics (Balazadeh et al., 2024; Chi et al., 2011). For example, a policy that improves the average learning outcomes of a group of students may not benefit every individual student equally. As a result, we introduce an approach, Human-centric Behavior-aware adaptive Off-policy selection (HBO), that can adapt the to the necessities of each participant given real-time observations over the task horizon.

### 3.1 PROBLEM FORMULATION

In HCSs, agents often operate with incomplete or noisy observations of user states, such as intentions and cognitive load, which are not fully observable or directly measurable. This partial observability naturally motivates modeling the practical problem as a partially observable Markov decision process (POMDP), where the agent could infer hidden states from observation histories to make informed decisions. We consider a setting defined by a POMDP, represented as a 7-tuple $(\mathcal{S}, \mathcal{A}, \mathcal{O}, \mathcal{T}, \mathcal{S}_0, \mathcal{R}, \mathcal{Z}, \gamma)$. Specifically, $\mathcal{S}$ is the state space which we assume is unknown, $\mathcal{A}$ is the action space, $\mathcal{O}$ is the observation space, $\mathcal{T} : \mathcal{S} \times \mathcal{A} \to \mathcal{S}$ defines transition dynamics from the current state and action to the next state, $\mathcal{S}_0$ defines the initial state distribution, $\mathcal{R} : \mathcal{S} \times \mathcal{A} \to \mathbb{R}$ is the reward function, $\mathcal{Z}(o|s)$ is the observation model which is unknown, $\gamma \in [0, 1)$ is discount factor. All episodes have a finite horizon $H$.

A trajectory under policy $\pi$ is denoted as $\tau_\pi^i = [\ldots, (o_t^i, a_t^i, r_t^i, o_{t+1}^i), \ldots]_{t=0}^H$. We have access to a historical (*i.e.*, offline dataset) collected under a behavioral policy $\beta$, $\mathcal{D}_\beta = \{..., \tau_\beta^i, ...\}_{i=1}^N$, which consist of $N$ trajectories where each trajectory corresponds to a single participant.

**Assumption 3.1** (Initial Policy Selection). Assume that when a new participant $i'$ joins the HCS, there always exist a pre-trained policy $\pi_1 \in \Pi$ assigned to $i'$ immediately upon the system receiving their initial observation $o_1^{i'}$; here, $\Pi$ is the set of policies, pre-trained on historical data $\mathcal{D}_\beta$, that have been approved by experts *a priori*. This initial policy selection is based on a *pre-given* mapping $\mathcal{D}_\beta \times \mathcal{O} \to \Pi$ to ensure a smooth start of the new participant (*e.g.*, determined by the experts). More details of initial policy selection in HCS are discussed in Appendix B.6.

Then, to maximize the outcome of each participant over the horizon, we aim to select the best sequence of policies $(\pi_1^*, ..., \pi_H^*)$ such that each policy $\pi_t^* \in \Pi$ is optimal at each step $t$, for each of the new participants $i' \in \mathcal{I}' = \{N+1, N+2, ...\}$ arriving at the system with an initial observation $o_1^{i'}$ (while the rest of the trajectory remains *unobservable*), that maximizes the participant's expected cumulative return, $V^{(\pi_0^*, ..., \pi_H^*)}$, $\max_{\pi_t^* \in \Pi, t \in [0,H]} V^{(\pi_0^*, ..., \pi_H^*)}$, over the full horizon $H$. Here $V^{(\pi_0^*, ..., \pi_H^*)} = \mathbb{E}_{(o_{t+1}, a_t) \sim \rho^{\pi_t^*}}[\sum_{t=0}^{H} \gamma^{t-1} r_t | (\pi_0^*, ..., \pi_H^*)]$, and $\rho^{\pi_t^*}$ is the observation-action visitation distribution under $\pi_t^*$ at step $t$.

However, motivated by the general guidance of *minimal trials*[1] in HCSs (Nie et al., 2021; Gao et al., 2024b), as well as to ensure sufficient bandwidth during online testing (*e.g.*, in randomized trials), in this work we consider identifying a single time point $h \in [1, H]$ at which a one-time policy assignment switch is made for each new participant $i' \in \mathcal{I}'$. The formal problem statement is below.

*Problem* 3.2 (Human-Centric Adaptive Off-Policy Selection Problem). The policy switch is expected to occur at most once over the horizon for each participant with minimal burden on participants. Specifically, the goal is that, given the fraction of the observed trajectory, $\tau_{\pi_{t<h}}^{i'}(0 : h) = [..., (o_t^i, a_t^i, r_t^i, o_{t+1}^i), ...]_{t=0}^h$, pertaining to a new participant $i'$ treated with the expert-selected policy $\pi_{t<h} \in \Pi$ (see Assumption 3.1) up until the current step $h \in [1, H]$, determine if switch to another policy $\pi_{t \geq h} \in \Pi \backslash \pi_{t<h}$ can maximize the total discounted sum of rewards, i.e., $\max_{h, \pi_{t \geq h}} \left( V^{\pi_{t<h}}(h) + V^{\pi_{t \geq h}}(h) \right)$; here, $V^{\pi_{t<h}}(h) = \mathbb{E}[\sum_{t=0}^{h-1} \gamma^t r_t | \pi_{t<h}, s_0]$ and $V^{\pi_{t \geq h}}(h) = \mathbb{E}[\sum_{t=h}^{H} \gamma^t r_t | \pi_{t \geq h}, \tau_{\pi_{t<h}}(0 : h), s_h]$ are the expected cumulative return before and after the policy switch at the time step $h$, respectively.

## 3.2 CRITICAL BEHAVIORAL PATTERNS (CBPs)

Historical data collected from HCSs in general provides limited coverage of the state/observation space, given the high-stake nature (Mandel et al., 2014). This limited coverage can make the data highly implicit, and hence unidentifiable for decision-making in its raw form; *e.g.*, patients appeared with similar current symptoms could be caused by different underlying diseases, where the clinicians will rely on their knowledge and past experience to carry out diagnoses and treatment plans onwards.

To tackle this challenge, in this section, we introduce a critical behavior patterns (CBPs) mining approach, to map observations into a discrete space; in what follows, sub-grouping can be leveraged to identify critical behavioral changes from the *historical* trajectories, which could be pivotal for HBO (see Algorithm 1) to leverage and decide if the policy assignments should be changed for any *new participants* at the current time step.

***Step I – Discrete Representation Mapping.*** Discrete representation mapping has proven effective in capturing general abstractions from complex trajectories and capturing similar behaviors shared across samples (Yang et al., 2021; Gao et al., 2024a). This is important for HCSs as often the observation space is large while offline coverage is low (Gao et al., 2023b). Specifically, each trajectory $\tau^i$ can be mapped to a temporal sequence $\chi^i$ of $H$ low-dimensional discrete representations, *i.e.*, $\chi^i = (z_0^i, z_1^i, ..., z_H^i)$, where $z_t^i$ is the representation at step $t$. This mapping could be facilitated by existing pattern mining algorithms, such as Toeplitz Inverse Covariance-Based Clustering (TICC) (Hallac et al., 2017) or Multi-series Time-aware TICC (MT-TICC) (Yang et al., 2021) which considers both cross-trajectory and temporal dependencies to apply discrete representation mapping.

***Step II – Identify CBPs.*** Then, sub-sequences of discrete representations $\chi_{t_1:t_2}^i$, with $0 \leq t_1 < t_2 \leq H$, are extracted from each $\chi^i$. CBPs are the sub-sequences of discrete representations that appear frequently in the corresponding trajectories with undesirable final outcome (*e.g.*, progression of pathology or failed final exams), which can indicate *critical behavioral changes* where more aggressive treatments may be necessary. The set of CBPs is denoted as $\tilde{\chi} = \{\chi_{t_1:t_2}^i | i \in [1, N]\}$.[2] More implementation details are provided in Appendix E.1.

---

[1] Refer to studies scoped to gather preliminary data on a new treatment or intervention with minimal burden on participants.

[2] Without loss of generality, $\chi_{t_1:t_2}^i$'s only refers to CBPs from this paragraph onwards. We slightly abuse the use this notation as the all other non-critical patterns will be discarded.

## 3.3 Confidence-Informed Policy Switch

Many human-centric practices in the real world benefit from the increased volume of acquired observations to boost confidence when making decisions (Dann et al., 2019; Nie et al., 2021), due to the partial observability in HCSs. Motivated by this pattern, here we introduce a confidence estimation approach that balances the benefits against opportunity costs of switching the policy at the current time step $h$, *i.e.*, if one should collect the return of switching to a (seemingly) more beneficial policy earlier (according to existing observations) or switch the policy later when more observations are available to be more confident.

For a new participant, $i'$, determining whether they will benefit from a policy switch at the current step $h$ depends on two factors, *i.e.*, (*i*) if CBPs are identified from the past observations (following Section 3.2); and (*ii*) if switching to $\pi_{t \geq h} = \arg\max_{\pi \in \Pi} \mathbb{E}[\sum_{t=h}^{H} \gamma^t r_t | \pi, \tau_{\pi_{t<h}}(0 : h), s_h]$ is estimated to result in more gains than sticking with $\pi_{t<h}$ and switching later once more observations are available.

**Who would benefit from policy switching.** Consider the trajectory $\tau_{\pi_{t<h}}^{i'}$ until step $h$, pertaining to the new participant $i'$ who is still treated by the expert-selected policy $\pi_{t<h}$. There exists a *switching advantage* by switching from $\pi_{t<h}$ to $\pi_{t \geq h} \in \Pi \backslash \pi_{t<h}$ when $V^{\pi_{t \geq h}}(h) > V^{\pi_{t<h}}(h)$. And we aim to determine who is likely to gain the switching advantage during the online deployment. As described in Section 3.2, CBPs are the patterns significantly presented in the *historical data* that had led to undesirable final outcomes. If the fraction of the observed trajectory, $\tau_{\pi_{t<h}}^{i'}(0 : h)$, of the new participant is identified as containing the CBPs, the participant has a risk to achieve undesirable final outcomes and may gain switching advantage.

To detect whether $\tau_{\pi_{t \leq h}}^{i'}(0 : h)$ contains CBPs, we first encode $\tau_{\pi_{t \leq h}}^{i'}(0 : h)$ into $\chi_{1:h}^{i'}$ using TICC-based methods, *i.e.*, by applying *Step I* of Section 3.2. Then if $\chi_{1:h}^{i'}$ contains any pre-identified CBPs from historical data (*i.e.*, in $\tilde{\chi}$), we flag this participant and keep tracking if switching to $\pi_{t \geq h}$ at current step $h$ or switching later once more observations are available, as introduced below.

**When to switch policy.** Given the interaction is performed in real time where observations onward from current step $h$ are unseen, it's not practical to wait until receiving entire trajectory of the participant and then decide the best step to switch policies. To address this, one can adopt off-policy selection to estimate the switching advantage by comparing estimated returns achieved by switching or not, *i.e.*, $\hat{V}^{\pi_{t \geq h}}(h)$ and $\hat{V}^{\pi_{t<h}}(h)$ respectively. That is also reminiscent of the decision-making process of human experts, by monitoring participants' behaviors and regularly re-evaluating whether to update interventions based on observations accumulated so far and if any abnormalities have already been detected (*i.e.*, the presence of CBPs). Specifically, the estimated value of switching to another policy, $\pi_{t \geq h} \in \Pi \backslash \pi_{t<h}$, is a variant of weighted importance sampling (WIS) (Precup et al., 2000) where only historical sub-trajectories containing the CBPs of interests are used. Specifically,

$$\hat{V}^{\pi_{t \geq h}}(h) = \frac{\sum_{i \in \Phi^{i'}} w(\tau^i, \pi_{t \geq h}) \sum_{t=h}^{H} \gamma^t r_t^i}{\sum_{i \in \Phi^{i'}} w(\tau^i, \pi_{t \geq h})}, \tag{1}$$

where $\Phi^{i'} \subseteq \{1, 2, ..., N\}$ is the subset of historical trajectory indices, whose corresponding discrete sequences contain the same identified CBPs from the current observations $\tau_{\pi_{t<h}}^{i'}(0 : h)$ of the new participant $i'$, $w(\tau^i, \pi_{(\cdot)}) = \prod_{t=h}^{H} \frac{\pi_{(\cdot)}(a_t | \tau_\beta^i(0:t))}{\beta(a_t | \tau_\beta^i(0:t))}$ is the importance ratio for the trajectory $\tau^i$ in the offline dataset. Similarly, $\hat{V}^{\pi_{t<h}}(h)$ can be calculated following Equation 1 by plugging $\pi_{t<h}$ into $w(\tau^i, \pi_{(\cdot)})$. Therefore, the optimal target policy for the new participant to switch from step $h$ can be obtained by

$$\pi_{t \geq h}^* = \underset{\pi \in \Pi \backslash \pi_{t<h}}{\arg\max} \hat{V}^\pi(h). \tag{2}$$

If $\hat{V}^{\pi_{t \geq h}^*}(h) > \hat{V}^{\pi_{t<h}}(h)$, there may exist an advantage to deploy $\pi_{t \geq h}^*$ to the participant $i'$ starting from the current step $h$. To better balance the gain achieved from switching immediately against the confidence from holding until more observations are collected, we introduce the *look ahead mechanism* to bound our confidence on switching at the current step. Specifically, once a CBP is detected and $\hat{V}^{\pi_{t \geq h}^*}(h) > \hat{V}^{\pi_{t<h}}(h)$, the look ahead mechanism steps in as follows.

**Definition 3.3** (Look Ahead Advantage). The look ahead advantage $\Delta(h)$ is defined as the difference of returns between deferring one more step followed by switching to the estimated best policy $\pi^*_{t \geq h+1}$ from step $h + 1$, and switching to estimated best policy $\pi^*_{t \geq h}$ from current step $h$, given $\tau^{i'}_{\pi_{t<h}}(0:h)$, *i.e.*,

$$\Delta(h) = \hat{V}^{\pi_{t<h}, \pi^*_{t \geq h+1}}(h+1) - \hat{V}^{\pi^*_{t \geq h}}(h), \tag{3}$$

where $\hat{V}^{\pi_{t<h}, \pi^*_{t \geq h+1}}(h+1) = \hat{r}_h + \gamma \sum_s p(s^{i'}_{h+1} | \tau^{i'}_{\pi_{t<h}}(0:h), \hat{o}^{i'}_{h+1}) \hat{V}^{\pi^*_{t \geq h+1}}(h+1)$. Specifically, $\hat{r}_h$ is the estimated reward received at current step $h$ by remaining on $\pi_{t<h}$, and $\hat{o}^{i'}_{h+1}$ is estimated observation by switching to policy $\pi^*_{t \geq h+1}$ at step $h + 1$. Both parts can be estimated through a model-based approach (Hafner et al., 2020; Gao et al., 2022); details are in Appendix E.2. Moreover, $p(s^{i'}_{h+1} | \cdot, \cdot)$ is the POMDP belief state model. As a result, if $\Delta(h) > 0$, the policy switch should be on-hold and deferred to the next step when there is no looking ahead advantage.

The overall HBO algorithm is summarized in Algorithm 1. Below we derive the upper-bound error of HBO-selected policies.

### 3.4 THEORY

Under a few assumptions, it is straightforward to see that our proposed algorithm will at least achieve the same value as the behavior policy minus epsilon.

**Assumption 3.4** (Full coverage). For all possible target policies $\{\pi_i\}^{i=E}_{i=1}$, $\pi_i(a|o) > 0 \rightarrow \pi_b(a|o) > 0, \forall o \in \mathcal{O}, a \in \mathcal{A}$ where $E$ is the cardinality of $\Pi$.

**Assumption 3.5** (In-distribution population). The distribution of trajectories in the original dataset $\tau \sim D$ is the same as any new trajectory $\tau'$.

**Assumption 3.6** ($\epsilon$-accurate policy value estimation). $|V^\pi - \hat{V}^\pi| \leq \epsilon$

**Proposition 3.7.** *Given 3.4, 3.5, 3.6, in each individual's MDP, the value of the selected policy $V^{HBO} \geq V^\beta - 2 * \epsilon$.*

*Proof.* Note that for all time steps at which HBO follows the behavior policy, the bound trivially holds. We therefore consider a subjtrajectory $t_h$ at which HBO

---

**Algorithm 1** HBO.

**Input:** A set of target policies $\Pi$, offline dataset $\mathcal{D}_\beta$. A pre-trained initial policy $\pi_1$. An operation policy $\pi_o$.
    // Offline Phase.
1: Get discrete sequences $\chi$ from $\mathcal{D}_\beta$.
2: Get the set of critical behavioral patterns (CBPs) $\tilde{\chi}$ from $\chi$ following Section 3.2.
    // Deployment Phase.
3: **while** the HCS receives the initial observation $o_1$ from a new participant **do**
4:    Initialize $\pi_o = \pi_1$.
5:    **for** each step $h \in [1, H]$ **do**
6:        Obtain the discrete representation $z_h$ of $o_h$.
7:        **if** $\tau_{\pi_{t \leq h}}(0:h)$ contains CBPs **then**
8:            Get estimated optimal switching policy $\pi^*_{t \geq h}(h)$ from Equation 2.
9:            **if** $\hat{V}^{\pi^*_{t \geq h}}(h) > \hat{V}^{\pi_{t<h}}(h)$ **then**
10:               Get look ahead advantage $\Delta(h)$ from Equation 3.
11:               **if** $\Delta(h) \leq 0$ **then**
12:                  Switch to $\pi^*_{t \geq h}(h)$. Update $\pi_o = \pi^*_{t \geq h}(h)$.
13:               **end if**
14:            **end if**
15:        **end if**
16:        Operate policy $\pi_o$.
17:    **end for**
18: **end while**

---

switches to another policy $\pi_i$, which it then follows for the remaining horizon. Consider the worst case setting in which HBO overestimates the value $\hat{V}^{\pi_i}(\tau_h)$ of following a given target policy $\pi_i$, and underestimates the value of the behavior policy $\hat{V}^{\pi_i}(\tau_h)$. $\pi_i$ is selected by HBO because $\hat{V}^{\pi_i}(\tau_h) > \hat{V}^\beta(\tau_h)$. By 3.6, this implies the following:

$$V^{\pi_i}(\tau_h) + \epsilon > \hat{V}^\beta(\tau_h) > \hat{V}^\beta(\tau_h) > \hat{V}^\beta(\tau_h) - \epsilon$$
$$V^{\pi_i}(\tau_h) > V^\beta(\tau_h) - 2 * \epsilon$$

$\square$

Note that the theory provides bounded error of HBO under given assumptions, and our empirical results provides additional evidence that HBO can still achieve superior performance even when some assumptions might not be perfectly met, *e.g.*, out-of-distribution scenarios as in Appendix B.

## 4 EXPERIMENTS

HBO is evaluated with two types of HCSs: interactive education and healthcare. Specifically, the interactive education experiment is based on a semi-synthetic simulator capturing the behavior of

1,342 students who participated in real-world college entry-level probability course over 7 academic semesters (Gao et al., 2022; 2024b; 2023b). The goal is to use the data collected from half of the students under an expert-designed policy, to assign pre-trained RL policies to the rest of students, in order to maximize their final learning outcomes. The healthcare experiment targets selecting pre-configured policies that can best treat patients with sepsis, over a simulated environment widely adopted in existing works (Namkoong et al., 2020; Tang & Wiens, 2021; Lorberbom et al., 2021; Gao et al., 2023a). More detailed experimental setups are provided in Appendices C, D, and F.

## 4.1 BASELINES

We now describe an additional variant of the HBO method we introduced, four baselines, and an oracle setting.

**(i) Vanilla one-policy-fits-all with existing OPE/OPS.** This baseline chooses a single target policy, $\pi \in \Pi$, to be deployed to all participants, which achieves the maximum expected return over the *entire offline dataset* as estimated using by existing OPE/OPS methods, *e.g.*, (Precup, 2000; Thomas & Brunskill, 2016; Le et al., 2019; Nachum et al., 2019a). However, as different OPE/OPS algorithms may converge to varied policy selections (Fu et al., 2021), we consider deploying each single target policy as baselines. **(ii) Combinatorial one-policy-fits-all with existing OPE/OPS.** Firstly, we exhaustively list all possible combinations of initial policies, target policies to switch to, over all steps. This leads to the space, $(\Pi \times \Pi)^H$, HBO is searching in equivalently. Then, this baseline uses an OPE method to select from this (much larger) set of policies. We consider two popular OPE estimators, weighted importance sampling (WIS) (Precup, 2000) and doubly robust (DR) (Jiang & Li, 2016), that have been widely used in OPE/OPS works, denoted as WIS* and DR*, respectively. Moreover, one could view WIS* and DR* as providing insights into the impact of pattern matching, since they excluded CBPs and performed OPS and switching by comparing the OPE estimates using all data (rather than only that which match in the pattern). **(iii) First-glance off-policy selection (FPS).** FPS assigns target policies to each new participant, at $t = 0$ when only $o_0$ is available, based on partitioning participants into sub-groups using both historical dataset and initial state of the new participant before online interactions (Gao et al., 2024b). **(iv) HBO with immediate policy switching (HBO-IS).** One variation of HBO is deciding policy switching once critical behavioral patterns are detected given a new participant, without looking ahead step introduced in Section 3.3. **(v) Oracle.** This approach computes the optimal selection for each participant in hindsight, *i.e.*, assuming at the beginning ($t = 0$) we could have picked for each participant the best policy as if the trajectory rollouts under all policies were available, what the final outcome would be. Although this approach is practically impossible as it would depend on access to future observations, it can inform approximately the best returns can be achieved by HBO (*i.e.*, the best policy is selected at $t = 1$).

## 4.2 INTELLIGENT TUTORING

The intelligent tutoring system (ITS) has been incorporated into an undergraduate-level course on probability and statistics across 7 semesters, involving a total of 1,342 students. The study was conducted with approval from the Institutional Review Board (IRB) at the institution, ensuring adherence to ethical standards. Additionally, a departmental committee oversees the process to safeguard participants' academic performance and privacy. In this educational setting, each learning session is structured with a sequence of 12 problems, referred to as an "episode" (horizon $H = 12$). During each step, the ITS provides students with three options: working independently, the students and the tutor co-construct the solution, or viewing the problem-solving demonstration from the ITS (mainly for studying purposes). The observation space per time steps consists of 142 log interaction features carefully designed by domain experts. These features capture various aspects of student activity, such as time spent on a problem and solution accuracy. There is zero reward for all intermediary steps. At the end of each episode, the reward is the normalized learning gain (NLG) derived from scores on two exams: a pre-test before using the system and a post-test afterward. The scores are normalized. Training data ($N = 628$) is collected under a lecturer-designed behavioral policy and used to train HBO for capturing critical behavioral patterns and estimating values of target policies. The initial behavior policy is the lecturer-ITS policy designed to ensure a smooth start and safety even without any policy switch. The target policies included two pre-trained alternate ITS RL policies. For additional details on NLG, pre-trained RL policies, and system setup, see Appendix C.

**Main results.** Figure 1 presents students' final outcomes under policies selected by different methods. Overall, HBO and its variations were the most effective policy selection methods, having the highest average students' final outcomes compared to other baselines. There was no obvious difference on the rounded averaged final returns across HBO and its variations, but we observed that a small set of individuals benefited from HBO compared to HBO-IS. This is discussed further in later sections. Moreover, HBO surpassed the state-of-the-art approach FPS, underscoring the efficacy of HBO in capturing students' underlying characteristics and switching policies with estimating confidence in real time. Though the traditional OPE/OPS methods can be adapted into the huge policy space (*i.e.*, overall $3^{12}$ combinations) as our algorithm searched in, both methods, WIS* and DR*, yielded lower final outcomes, further indicating the need for HBO. It is also noted that the performance of the initial policy was worse than other target policies (*i.e.*, RL-induced Policy 1 & 2), while

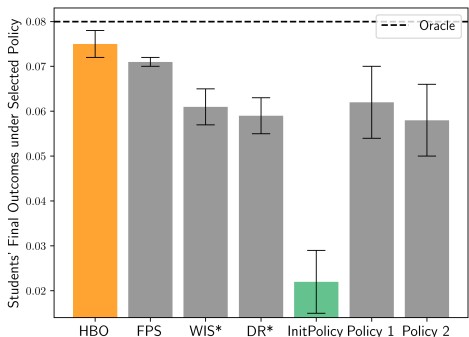

Figure 1: Final outcomes (*i.e.*, normalized learning gains) of students (mean±se) under selected policies in the testing period. WIS* and DR* are using WIS and DR for exhaustive switching combinations of target policies, respectively; InitPolicy is the lecturer-designed policy used for initiating interactions; Policy 1 & 2 are two RL-induced one-policy-fits-all policies.

HBO can significantly enhance students' performance beginning with the initial policies, highlighting HBO's significant advantage in correctly switching to the right policy at proper timing. Last but not least, our approach is getting quite close to the oracle's performance, further justifying the need for and advantages of HBO.

**Individual gains from looking ahead.** Figure 2 illustrates the distribution of final outcomes over lower-performers whose normalized learning gains were negative, under initial policy and HBO, respectively. It was observed that under the initial policy, a substantial portion of the cohort, specifically 229 out of 628 students, experienced negative learning gains. This high incidence of unfavorable outcomes highlights the limitations of the initial policy in adequately addressing the diverse needs of the student population. Conversely, under HBO, the number of students with negative learning gains significantly decreased to only 47 out of 628. This stark reduction by approximately 79% underscores HBO's capability to customize to support those who are failing to thrive under a one-size-fits-all default policy.

Though there was no obvious difference on the rounded averaged final returns across HBO and its

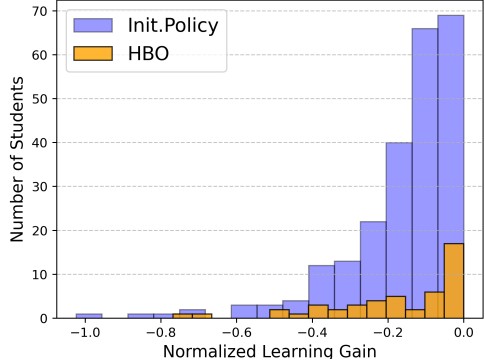

Figure 2: Histogram of the distribution of *lower-performers'* (*i.e.*, students got negative normalized learning gains) final outcomes. The x-axis represents the range of final scores, while the y-axis indicates the number of students within each score range.

variation HBO-IS, we observed that a small set of individuals benefited from HBO compared to HBO-IS (no delayed lookahead). The efficacy of HBO's personalized approach is further highlighted through detailed case studies which suggest a benefit of the one-step lookahead. A simulator captured participants' trajectories under corresponding policies, respectively, using variational auto-encoder as in Gao et al. (2024a; 2023b; 2024b), and rolled out sub-trajectories to simulate the continuing interactions from switching policies given current observations. In one case, a student's learning trajectory was analyzed for optimal policy switching points. While a simulated switch to an optimal policy at $t = 3$ predicted a return of 0.135, HBO postponed the switch to $t = 4$, resulting in a significantly higher return of 0.23. Similarly, another student's experience further justified this benefit brought by HBO. Specifically, immediate policy switching at $t = 3$ would yield a return of 0.15. However, by delaying the switch to $t = 4$ and $t = 5$, the returns progressively increased to 0.233 and 0.3, respectively. These case studies illustrate how HBO's nuanced understanding of individual learning patterns enables it to outperform standard approaches by dynamically adjusting policies not just to the learner's immediate needs, but also to their evolving potential over time, thereby enhancing final outcomes.

## 4.3 SEPSIS TREATMENT

We consider selecting the policy that can best treat sepsis for each patient in the ICU, leveraging the simulated environment introduced by Oberst & Sontag (2019), which has been widely adopted in existing works (Hao et al., 2021; Nie et al., 2022; Tang & Wiens, 2021; Lorberbom et al., 2021; Gao et al., 2023a; Namkoong et al., 2020). Specifically, the state space is constituted by a binary indicator for diabetes, and four vital signs {heart rate, blood pressure, oxygen concentration, glucose level} that take values in a subset of {very high, high, normal, low, very low}; size of the state space is $|\mathcal{S}| = 1440$. Actions are captured by combinations of three binary treatment options, {antibiotics, vasopressors, mechanical ventilation}, which lead to $|\mathcal{A}| = 2^3$. Seven candidate target policies are considered (Namkoong et al., 2020), *i.e.*, $(i)$ initial policy which is a soft physician policy, with 0.95 probability takes the action recommended by an RL policy trained following policy iteration, and with 0.05 probability picks randomly from other actions; $(ii)$ with antibiotics on the initial step which register antibiotics right after the patient is admitted; $(iii)$ without antibiotics on the initial step which does not administer antibiotics right after the patient is admitted; $(iv)$ a mixture of two policies: 80% of the initial policy and 20% of a policy that is similar to the initial policy but the vasopressors action is flipped; $(v)$ a mixture of two policies similar to $(iv)$ but 60% of the initial policy; $(vi)$ a policy that always administers antibiotics once the patient is admitted; $(vii)$ a policy that never administer antibiotics. Moreover, a simulated unrecorded comorbidities is applied to the cohort, capturing the uncertainties caused by patient's underlying diseases (or other characteristics), which could reduce the effects of the antibiotics being administered. See Appendix D for more details in regards to the environmental setup.

**Main results.** HBO consistently delivered significantly improved outcomes in the longer-horizon setup (*i.e.*, $H = 50$), and outperformed FPS and other baselines over all combinations of horizon and training size setups, which demonstrated its robustness to varied environments. In contrast, HBO-IS, which lacks the looking ahead mechanism and makes immediate policy decisions based on critical behavioral patterns, led to worse outcomes than HBO in all setups. It is also noted that the performance gap between HBO and HBO-IS is closer when $H = 20$, HBO-IS falls short more significantly with longer horizon, indicating that HBO-IS lacked capability in handling complex, evolving healthcare dynamics without the benefit of foresight.

Table 1: The final outcomes from deploying to each patient the corresponding candidate policy selected by HBO against baselines, on $H = \{20, 50\}$, with varied training size $N = \{1,000, 5,000\}$, respectively. Results are averaged over 5 different runs. Standard errors are rounded. WIS* and DR* perform the same rounded averaged return as other baselines.

|  |  | H=20 | H=50 |
|---|---|---|---|
| One-policy-fits-all | Init. Policy | 0.068 (.007) | 0.025 (.007) |
| | With Antibiotics Init | 0.068 (.000) | 0.021 (.000) |
| | Without Antibiotics Init | 0.037 (.000) | -0.02 (.000) |
| | 80% Mix w. Expert | 0.005 (.000) | -0.06 (.000) |
| | 60% Mix w. Expert | -0.022 (.000) | -0.1 (.000) |
| | With Antibiotics All-way | 0.044 (.000) | 0.022 (.000) |
| | Without Antibiotics All-way | 0.012 (.000) | -0.015 (.000) |
| N=1,000 | FPS | 0.08 (.003) | 0.026 (.004) |
| | HBO-IS | 0.081 (.007) | 0.034 (.007) |
| | HBO | 0.086 (.007) | **0.039** (.002) |
| N=5,000 | FPS | 0.07 (.003) | 0.032 (.003) |
| | HBO-IS | 0.079 (.002) | 0.033 (.002) |
| | HBO | **0.081** (.002) | **0.039** (.001) |
| | Oracle | 0.166 (.008) | 0.238 (.007) |

## 5 CONCLUSION, LIMITATION, AND FUTURE WORK

In this work, we introduced HBO which provides adaptive personalized policy selection by capturing potentially unique characteristics from human behaviors and estimates when and how to intervene with less uncertainty in a timely manner. HBO was validated with extensive real-world human-centric applications, intelligent tutoring and sepsis treatment, where HBO achieved superior performance compared to baselines. While this study primarily focuses on the off-policy selection task to identify improvements from capturing underlying unique human characteristics and adaptive policy assignments with minimal trials, further work could include extending HBO for full policy optimization beyond policy selection. Also, our assumption 3.4 may hold in many human-centric domains (such as our educational tutoring data and sepsis treatment) the deployed behavior policies are relatively consistent (e.g., due to standardized curricula and treatment protocols). If overlap is highly limited, one could use more conservative evaluation techniques. Relaxing or adapting this assumption is an important direction for future work. Future work may involve developing policy estimators that manage the bias-variance trade-off, such as incorporating weighted doubly robust or MAGIC methods (Thomas & Brunskill, 2016) instead of importance sampling weights.

Incorporating a budgeted multi-switch model could increase flexibility while still managing overall risk in lower-risk scenarios such as recommendation systems. Moreover, our current switching rules based on CBPs are deliberately conservative to ensure safety: we only switch when there is strong evidence of a problematic pattern (as this is necessary in high-stake human-centric domains). A learned scoring model (for example, predicting expected improvement) could allow more nuanced decisions, but it would require additional data and might risk overfitting or losing transparency. High-stakes environments, such as healthcare, can restrict the range of available policy options. Our results suggest HBO is a promising approach for such settings, as it is an effective and scalable method that can work well even with only relatively small offline datasets. Limitation is discussed in Appendix A.1.

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

LIST OF APPENDICES

## A  Limitation and Broader Impacts

### A.1  Limitation

Limitations of this work includes ($i$) the semi-synthetic sepsis simulator lacks capabilities in capturing side effects of aggressive treatments (*i.e.*, to penalize the use of overly aggressive procedure at early stages), which resulted in the significant gap between Oracle's performance against all other methods considered. On the other hand, this also demonstrated the practical challenges in the healthcare domain, *i.e.*, the right decisions need to be made timely within a short window alongside high degree on uncertainties. This is especially true for acute diseases, where balancing the risk of disease progression against the side effects of overly aggressive treatments is particularly challenging. ($ii$) In future work, HBO can be also extended to explicitly identify and separate the potential complications from multiple sources, *e.g.*, cancer patients are often affected by weakened immune systems, making them more sensitive to germs and viruses that are not harmful to normal people, or low performer students who are also going through difficult times due to family or other personal reasons.

### A.2  Broader Impacts in RL-Empowered HCSs

All real-world data employed in this paper were obtained anonymously through exempt IRB-approved protocols and were scored using established rubrics. No demographic data or class grades were collected. All data were shared within the research group under IRB, and were de-identified and automatically processed for labeling. This research seeks to remove societal harms that come from lower engagement and retention of students who need more personalized interventions and developing more robust medical interventions for patients.

Fairness in AI-empowered HCSs has been a long-standing concern (Lepri et al., 2021; Metevier et al., 2019; Nie et al., 2023; Ruan et al., 2023; Elmalaki, 2021). The HBO framework can be potentially extended to promote fairness to a certain extent, by helping minority/under-represented groups to boost their utility gain through deployment of customized policy specific to the group. Specifically, following HBO, the critical behavior patterns can identify the small-scaled yet important groups, then the policy that is most beneficial for the group can be deployed to maximize the gain. As illustrated by Figure 2, HBO effectively boosts low-performers' final outcomes with significantly reduced numbers of lower-performers compared to deploying the initial policy. Similarly, one can easily extend the HBO framework to intelligent HCSs oriented toward other applications, in order to identify the groups that potentially need more attention, and help all participants to achieve similar gain indiscriminately by deploying the right policy to each participant.

## B  Additional Experimental Investigation and Discussion

### B.1  Out-of-Distribution Scenario

We made the assumption for general purposes that training data may cover the distribution of population (*e.g.*, patients visiting a hospital may mainly come from the same area). But we agree that out-of-the-distribution is also a challenging and interesting direction in practice. To help explore this, we added a sepsis simulation on an out-of-distribution setting where we changed the probability of having diabetes to 0.1 at test time (during training it was 0.2 by default), which impacts the dynamics of the system – note diabetes is an unobservable state feature (Oberst & Sontag, 2019). For horizon is 50 and training size is 5,000, HBO achieved 0.042 (.007), better than baselines 0.038 (.003), 0.039 (.008), 0.036 (.008), by FPS, the best one-policy-fits-all, the best combinatorial one-policy-fits-all, respectively. The oracle reached 0.571 (.006). Results were averaged over 5 different simulation runs.

### B.2  Unlimited/Multiple Switching

One might imagine that conceptually multiple switches may lead to better optimality, however, practically there may not be much opportunities for trial-and-error given the high-stake and highly regulated nature of HCSs; there may be logistic considerations or other constraints (doctors might not want patients to be exposed to many different chemotherapy drugs). In sepsis, we found that the switching multiple times led to an average return 0.037 (.000) for sepsis (H=50, N=5,000), slightly

smaller than HBO (0.039 (.001)). But note that multiple switching may raise safety concerns by stakeholders, we agree it would be a promising direction especially for lower-stake scenarios such as recommended systems and human-computer interaction.

## B.3 THE CONTRIBUTION OF CBPS

As discussed in Section 4, one could view our baselines WIS* and DR* as providing insights into the impact of pattern matching, since they excluded pattern mining and performed off-policy selection and switching by comparing the OPE estimates using all data (rather than only that which match in the pattern). As our method HBO outperforms both baselines, this suggests that our approach benefited from pattern mining in general. To further investigate the sensitivity of potentially poor critical-pattern mining and matching, we randomly select patterns from the large set of candidates for extracting the subset of data used for performing the off policy evaluation. This led to an average return of 0.031 (.005) for sepsis treatment (H=50, N=5,000) from 5 different runs, smaller than HBO (0.039 (.001)).

## B.4 SMALL DATA SIZES

We ran additional trials by subsampling our datasets to 10% of the original size. For horizon is 50 and both training and test sizes are 500 in sepsis treatment, HBO achieved 0.038 (.023), better than baselines 0.027 (.012), 0.017 (.024), 0.02 (.02), by FPS, the best one-policy-fits-all, the best combinatorial one-policy-fits-all, respectively. Results were averaged over 5 different simulation runs. Even with this limited data, HBO continued to outperform baseline methods, although performance variance increased as expected.

## B.5 REAL-WORLD CBPS SPECIFICS

It could be hard to explicitly interpret the identified CBPs, given they were extracted from low-dimensional representations encoded from high-dimensional data, and each individual can have varied histories before the pattern occurred. We investigated the identified CBPs on some easy-to-interpret features in intelligent tutoring, such as the chance hints are requested for a problem, average time spent for each step, and answer correctness for each problem. Table 2 shows the mean values on the three features associated with a CBP (length=2) that was more frequent in lower performers ($\sim 10\%$ more frequent than it occurred among higher performers). Results show that the CBP is associated with the participants requesting less hints but spending more time on answering, and their correctness was relatively higher than average but exhibited a tendency to decrease.

Table 2: Some CBPs Specifics in Intelligent Tutoring.

| AvgValue | HintRates | TimeSpent | Correctness(%) |
|---|---|---|---|
| Across all steps and participants | 0.006 | 0.031 | 83 |
| 1st state element of the CBP | 0.001 | 0.046 | 91 |
| 2nd state element of the CBP | 0.002 | 0.050 | 88 |

## B.6 INITIAL POLICY IN ASSUMPTION 3.1

We assume the initial policy is always safe but could be suboptimal (*e.g.*, determined by domain experts and passed the sanity check overseen by related departments). For example, in education, we follow an instructor-defined policy which has been shown to be effective on student learning outcomes, but it is less effective compared to the best of the RL-induced policies for specific individuals. The expert policy is usually maximizing the expectation of the entire cohort (the good outcomes could hardly become better but the worse outcomes don't get too worse), while our work makes the treatments more personalized. In sepsis treatment, the initial policy could be an $\epsilon$-variant of a policy which takes actions from a RL policy learnt from historical data or physician policy but with a small probability picked randomly from other actions (Namkoong et al., 2020). In our experiments, we observe $\sim 10\%$ times the initial expert policy is selected as the optimal policy in education, and

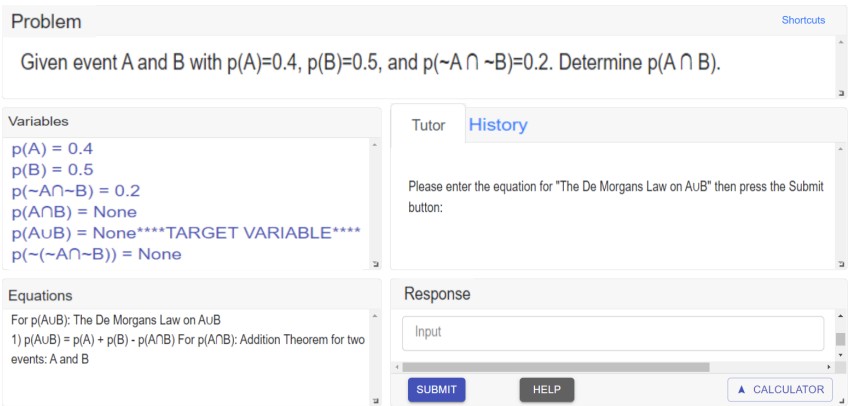

Figure 3: Graphical user interface (GUI) of the IE system. The problem statement window (*top*) presents the statement of the problem. The dialog window (*middle right*) shows the message the tutor provides to the students. Responses, e.g., writing an equation, are entered in the response window (*bottom right*). Any variables and equations generated through this process are shown on the variable window (*middle left*) and equation window (*bottom left*).

$\sim 39\%$ times (averaged over 5 runs) for patients in sepsis treatments (H=50, N=5,000), probably because the expert policy is a near optimal one as noted by (Namkoong et al., 2020).

**Discussion on cold-start case when no expert-approved policies are available.** Assumption 3.1 is common in high-stake human-centered domains: for example, a standard curriculum or default intervention often exists by design. If no expert policy is available (usually may happen in a lower-stake task), one could use a very conservative fallback (e.g., continuing at the slowest pace) as an initial safe policy. Alternatively, the framework could bootstrap a safe policy by imitating minimal data or by adapting from a known baseline.

## C  INTELLIGENT TUTORING SETUP

### C.1  THE INTERACTIVE EDUCATION (IE) SYSTEM FOR THE COLLEGE ENTRY-LEVEL COURSE.

To validate the proposed method in real-world human-centric applications, we conduct experiments with data collected on the system specifically used in an undergraduate probability course at a university, which has been extensively used by over $1,342$ students with $\sim$860k recorded interaction logs through 7 academic semesters. The IE system is designed to teach entry-level undergraduate students with ten major probability principles, including complement theorem, mutually exclusive theorem, independent events, De Morgan's theorem, addition theorem for two events, addition theorem for three events, conditional independent events, conditional probability, total probability theorem, and Bayes' rule.

Each students went through four phases, including (*i*) reading the textbook; (*ii*) pre-exam; (*iii*) studying on the IE system; and (*iv*) post-exam. During the reading textbook phase, students read a general description of each principle, review examples, and solve some training problems to get familiar with the IE system. Subsequently, they take a pre-exam comprising a total of 14 single- and multiple-principle problems. During the pre-exam, students are not provided with feedback on their answers, nor are they allowed to go back to earlier questions (so as the post-exam). Then, students proceed to work on the IE system, where they receive the same 12 problems in a predetermined order. After that, students take the 20-problem post-exam, where 14 of the problems are isomorphic to the pre-exam and the remainders are non-isomorphic multiple-principle problems. Exams are auto-graded following the same grading criteria set by course instructors.

Since students' underlying characteristics and mind states are inherently unobservable (Mandel et al., 2014), the IE system defined its state space with 142 features that could possibly capture students' learning status based on their interaction logs, as suggested by domain experts. While

tutoring, the agent makes decisions on two levels of granularity: problem-level first and then step-level. For problem-level, it first decides whether the next problem should be a worked example (`WE`) (Sweller & Cooper, 1985), problem-solving (`PS`), or a collaborative problem-solving worked example (`CPS`) (Schwonke et al., 2009). In `WE`s, students, observe how the tutor solves a problem; in `PS`s, students solve the problem themselves; in `CPS`s, the students and the tutor co-construct the solution. If a `CPS` is selected, the tutor will then make step-level decisions on whether to `elicit` the next step from the student or to `tell` the solution step to the student directly. Besides post-exam score, another important measure of student learning outcomes is their normalized learning gain (NLG), which is calculated by their pre- and post-exam scores $NLG = \frac{score_{postexam} - score_{preexam}}{\sqrt{1 - score_{preexam}}}$.

The NLG defined in (Chi et al., 2011), represents the extent to which students have benefited from the IE system in terms of improving their learning outcomes.

## C.2   CLASSROOM SETUP

**Participants recruitment.** All participants were entry-level undergraduates majoring in STEM and enrolled in the Probability course in a college. They were recruited via invitation emails and told the procedure of the study and their data were used for research purpose only, and the study was an opt-in without influence on their course grades. Participants can also opt-in not recording their logs and quit the study any time. No demographics data or course grades were collected. All participants had acknowledged the study procedure and future research conducted using their logs.

**Principles taught by the IE system.** Table 3 shows all ten principles for the IE system to teach designed for the undergraduate entry-level students with STEM majors.

**Pre- and post-exams.** We use pre- and post-exams to measure the extent to which students have benefited from the IE system for improved learning outcomes. Tables 4 & 5 contain all problems in pre- and post-exams during our experiment with the IE system.

**The set of candidate target policies under consideration.** For safety, two RL-induced target policies that passed expert sanity checks can be deployed in one semester, while the expert policy still remained in each semester as the control group. For fairness concerns, the IE system randomly assigned a policy to each student. Overall, 1256, 42, and 44 students accomplished all problems and exams, who were assigned the expert policy, RL-induced policy 1, and RL-induced policy 2.

Table 3: Principles taught by the IE system for undergraduate entry-level students.

| Abbr. | Name of principle | Expression |
|-------|-------------------|------------|
| CT | Complement Theorem | $P(A) + P(\neg A) = 1$ |
| MET | Mutually Exclusive Theorem | $P(A \cap B) = 0$ iff A and B are mutually exclusive events |
| IE | Independent Events | $P(A \cap B) = P(A)P(B)$ if A and B are independent events |
| DMT | De Morgan's Theorem | $P(\neg(A \cup B)) = P(\neg A \cap \neg B), P(\neg(A \cap B)) = P(\neg A \cup \neg B)$ |
| A2 | Addition Theorem for two events | $P(A \cup B) = P(A) + P(B) - P(A \cap B)$ |
| A3 | Addition Theorem for three events | $P(A \cup B \cup C) = P(A) + P(B) + P(C) - P(A \cap B) - P(A \cap C) - p(B \cap C) + P(A \cap B \cap C)$ |
| CIE | Conditional Independent Events | $P(A \cap B|C) = P(A|C)P(B|C)$ if A and B are independent events given C |
| CP | Conditional Probability | $P(A \cap B) = P(A|B)P(B) = P(B|A)P(A)$ |
| TPT | Total Probability Theorem | $P(A) = P(A|B_1)P(B_1) + P(A|B_2)P(B_2) + \ldots + P(A|B_n)P(B_n)$ |
| | | if $B_1, B_2, \cdots, B_n$ are mutually exclusive events and $B_1 \cup B_2 \cup \cdots \cup B_n = W$ |
| BR | Bayes Rule | $P(B_i|A) = P(A|B_i)P(B_i)/P(A|B_1)P(B_1) + P(A|B_2)P(B_2) + \cdots + P(A|B_n)P(B_n)$ |
| | | if $B_1, B_2, \cdots, B_n$ are mutually exclusive events and $B_1 \cup B_2 \cup \cdots \cup B_n = W$ |

Table 4: Pre-exam problems in the IE system.

| Problem | CT | MET | IE | DMT | A2 | A3 | CIE | CP | TPT | BR |
|---------|----|----|----|----|----|----|----|----|----|----|
| 1 | | | | | | ✓ | | | | |
| 2 | | | | | | | | ✓ | | |
| 3 | | | | | | | ✓ | | | |
| 4 | ✓ | | | ✓ | ✓ | | | | | |
| 5 | | | | | | | | | | ✓ |
| 6 | | | | | | | ✓ | | | ✓ |
| 7 | | ✓ | ✓ | | | ✓ | | | | |
| 8 | ✓ | ✓ | | ✓ | | ✓ | ✓ | | | |
| 9 | ✓ | | | | | | | | | |
| 10 | | | | | | ✓ | | | | |
| 11 | | | ✓ | | | | | | | |
| 12 | | | | | ✓ | | | | | |
| 13 | | | | | | | ✓ | | | |
| 14 | | ✓ | | | | | | | | |

### C.3 ENVIRONMENTAL SETUP OF THE IE SYSTEM

#### C.3.1 STATE FEATURES.

The state features were defined by domain experts that could possible capture students' learning status based on their interaction logs. In sum, 142 features with both discrete and continuous values are extracted, we provide summary descriptions of the features characterized by their systematic functions: (*i*) Autonomy (10 features): the amount of work done by the student, such as the number of times the student restarted a problem; (*ii*) Temporal Situation (29 features): the time-related information about the work process, such as average time per step; (*iii*) Problem-Solving (35 features): information about the current problem-solving context, such as problem difficulty; (*iv*) Performance (57 features): information about the student's performance during problem-solving, such as percentage of correct entries; (*v*) Hints (11 features): information about the student's hint usage, such as the total number of hints requested.

Table 5: Post-exam problems in the IE system.

| Problem | CT | MET | IE | DMT | A2 | A3 | CIE | CP | TPT | BR | Iso-Test-Problem |
|---------|----|----|----|----|----|----|----|----|----|----|------------------|
| 1 | | | ✓ | | | | | | | | 11 |
| 2 | | ✓ | ✓ | | | ✓ | | | | | 7 |
| 3 | ✓ | | | ✓ | ✓ | | | | | | 4 |
| 4 | ✓ | | | | | | | | | | 9 |
| 5 | | | | | | | | ✓ | | | 3 |
| 6 | | | | | | ✓ | | | | | 10 |
| 7 | | | | | | | | ✓ | | | 2 |
| 8 | | | | | | | ✓ | | | | 13 |
| 9 | | | | | | | | ✓ | ✓ | ✓ | N/A |
| 10 | | ✓ | | | | | | | | | 14 |
| 11 | | | | | | | | | | ✓ | 5 |
| 12 | | | | | ✓ | | | | | | 12 |
| 13 | | | | | | | ✓ | | | ✓ | 6 |
| 14 | ✓ | | ✓ | | | | | | | | N/A |
| 15 | ✓ | | | ✓ | ✓ | | | ✓ | | | N/A |
| 16 | ✓ | | | | | | | | | ✓ | N/A |
| 17 | | | | | | ✓ | | | | | 1 |
| 18 | ✓ | ✓ | | ✓ | ✓ | ✓ | | | | | 8 |
| 19 | ✓ | ✓ | | | ✓ | ✓ | | | | | N/A |
| 20 | | ✓ | | | | ✓ | | | ✓ | | N/A |

### C.3.2 ACTIONS & REWARDS.

See C.1 above.

### C.3.3 BEHAVIOR POLICY.

The behavior policy follows an expert policy commonly used in e-learning (Zhou et al., 2019), randomly taking the next problem as a worked example (`WE`), problem-solving by students (`PS`), or a collaborative problem-solving working examples (`CPS`). Note that the three decision choices are designed by domain experts that are found can support students' learning in prior works (Schwonke et al., 2009; Sweller & Cooper, 1985), thus the expert policy is considered as effective.

### C.3.4 TARGET (EVALUATION) POLICIES.

In total, three target policies, including two RL-induced and the expert policy, were examined. The RL-induced policies were trained using off-policy DQN-based algorithm, and passed expert sanity check. In this study, expert sanity check were conducted by departments and independent instructors for pre-examination of the target policies.

Specifically, we employed the DQN-based algorithm designed by domain researchers (Ju, 2019), called Critical-RL, that have achieved empirical significance in real-world classrooms, and passed expert sanity check by our institutions. In this study, we examined two variations of the Critical-RL, *i.e.*, a policy carrying out original policy when a decision is *not* critical, and a policy carrying out original policy over all decisions. We set the threshold to be the median Q-value difference for all decisions in our training data set following the settings of the original Critical-RL work (Ju, 2019). Each pair of of adversarial policies considered all parts of the training data were identical, such as state representation and transition samples, except the rewards. We use the learning rate $lr = 1e - 3$ for inducing DQN policies.

### C.4 DETAILED EXPERIMENTAL DESIGN TO VALIDATE PROPOSED METHOD

Since the study was under strict protocol control of IRB and required carefully fairness over equity and ethnicity, policies were deployed to each students without online switching. Since the initial policy would be the expert policy, and an RL-induced policy may be switched to during the interactions, to mimic realistic interactions from switching, we conducted semi-synthetic experiments which combined both real data, and simulated data generated by variational auto-encoders (VAEs) (Gao et al., 2022) that modeling the dynamics of student-system interactions under corresponding policies. Specifically, we randomly divided students under the expert policy into two groups, then used one group as historical data for training our proposed methods and baselines, while used the other group as the validation set. If a student in validation set was identified for policy switching, we deployed the simulator to mimic their interactive logs following the new policy, by rolling out sub-trajectories from the pre-trained VAE and choosing the sub-trajectory whose observations from the switch-beginning step is closest to the real observations at that step. The distance between two observations were calculated by Euclidean distance. Structures of VAEs and the hyper-parameters are provided in Appendix F.3. If a student in validation set was identified as no need of policy switching, then their original trajectory was kept for calculating final outcomes.

Baselines may use different portions of synthetic data during validation. Vanilla one-policy-fits-all with existing OPE/OPS purely used real validation data since the true returns under each target policy can be extracted. Combinatorial one-policy-fits-all with existing OPE/OPS purely used synthetic data, since they selected combined policies. FPS used real validation data since it determined the policy assignment before online interactions.

## D SEPSIS TREATMENT SETUP

We use the sepsis simulator developed by Oberst & Sontag (2019) and benchmark settings of (Namkoong et al., 2020). To construct the POMDP environment, data was generated with a small amount of confounding with $\Gamma = 2.0$ as in (Namkoong et al., 2020), which was unobservable for off-policy selection methods.

### D.1 STATES & ACTIONS.

The definition of states and actions are introduced in Section 4.3.

### D.2 REWARDS.

We also follow the benchmark (Namkoong et al., 2020) in terms of configuring the reward function and behavioral policy. Specifically, a reward of -1 is received at the end of horizon if the patient is deceased (*i.e.*, at least three vitals are out of the normal range), or +1 if discharged (when all vital signs are in the normal range without treatment).

### D.3 BEHAVIOR/INITIAL POLICY.

It used policy iteration to learn the optimal policy, and created a near-optimal (soft optimal) policy by having the policy take a random action with probability 0.05, and the optimal action with probability 0.95 (Namkoong et al., 2020). The value function (for the optimal policy) was computed using value iteration. The discount factor $\gamma = 0.99$.

### D.4 TARGET POLICIES.

See Section 4.3.

## E MORE ON METHODOLOGY

### E.1 LEARNING DISCRETE REPRESENTATIONS

First, observations $o_t^i$ from historical data $\mathcal{D}_\beta$ are mapped into $K$ clusters based on the values of observation variables, where each $o_t^i$ is associated with a corresponding cluster from the set $\mathbf{M} = \{M_1, ..., M_K\}$. $K$ can be greatly smaller than $H \times N$ as considered in general discrete representation mapping problems (Hallac et al., 2017; Yang et al., 2021). Then we extract sub-sequences of the latent representations with varied length in $[2, H]$ from latent trajectories $\chi^i, i \in [1, N]$ learnt from historical data, *i.e.*, $\chi_{h_1, h_2}^i = (z_{h_1}^i, ..., z_{h_2}^i), h_1 < h_2, h_1, h_2 \in [1, H]$ is a sub-sequence of $\chi^i$. We present using an straightforward off-the-shelf technique, TICC (Hallac et al., 2017), to encode trajectories into latent representations that considers both cross-trajectory and temporal dependencies in implementation.

To capture critical human behaviors that may indicate undesirable final outcomes under $\pi_1$ and need careful policy selection, we extract sub-sequences of latent representations using their mapping trajectories' outcomes from historical data, such that the sub-sequences of latent representations that are more frequent in sub-groups with worse final outcome (*e.g.*, deceased patients, students failed in a course) may indicate critical underlying states that require timely interventions. Specifically, it is possible that a temporal discrete sub-sequence $\chi_{h_1, h_2}^i$ is "equal" to another temporal discrete sub-sequence $\chi_{h_1, h_2}^j$, such that $\chi_{h_1, h_2}^i = \chi_{h_1, h_2}^j$ if every $z_{h_t}^i = z_{h_t}^j$. The frequency of any $\chi_{h_1, h_2}^i$ in historical data can be calculated by counting the number of historical trajectories whose associated discrete sequences contain the discrete sub-sequence: $frequency_\chi(\chi_{h_1, h_2}^i) = \sum_{j=1}^N (\mathbb{1}(\chi_{h_1, h_2}^i = \chi_{h_1, h_2}^j))$, where $\chi = \{\chi^i\}_{i=1}^N$ is the set of all extracted sub-sequences of latent representations from historical data $\mathcal{D}_\beta$, $\mathbb{1}(\cdot)$ is the indicator function. A $\chi_{h_1, h_2}^i$ is labeled as "critical behavior" if $frequency_\chi(\chi_{h_1, h_2}^i) \geq \epsilon_b$. The final set of critical behavior-labeled sub-sequences is denoted as $\tilde{\chi}$. In implementation, we extracted the top 10 critical behavioral patterns by sorting by their frequencies in descending order to form $\tilde{\chi}$.

### E.1.1 TICC PROBLEM

Each cluster $k \in [1, K]$ is defined as a Markov random field (Rue & Held, 2005), or correlation network, captured by its Gaussian inverse covariance matrix $\Sigma_k^{-1} \in \mathbb{R}^{c \times c}$, where $c$ is the dimension of state space. We also define the set of clusters $\mathbf{M} = \{M_1, \ldots, M_K\} \subset \mathbb{R}$ as well as the set of inverse covariance matrices $\mathbf{\Sigma}^{-1} = \{\Sigma_1^{-1}, \ldots, \Sigma_K^{-1}\}$. Then the objective is set to

be: $\max_{\mathbf{\Sigma}^{-1}, \mathbf{M}} \sum_{k=1}^{K} \left[ \sum_{o_t^i \in M_k} \left( \mathcal{L}(o_t^i; \Sigma_k^{-1}) - \sigma \mathbb{1}\{o_{t-1}^i \notin M_k\} \right) \right]$, where the first term defines the log-likelihood of $o_t^i$ coming from $M_k$ as $\mathcal{L}(o_t^i; \Sigma_k^{-1}) = -\frac{1}{2}(o_t^i - \mu_k m)^T \Sigma_k^{-1}(o_t^i - \mu_k) + \frac{1}{2} \log \det \Sigma_k^{-1} - \frac{n}{2} \log(2\pi)$ with $\mu_k$ being the empirical mean of cluster $M_k$, the second term $\mathbb{1}\{o_{t-1}^i \notin M_k\}$ penalizes the adjacent events that are not assigned to the same cluster and $\sigma$ is a constant balancing off the scale of the two terms. This optimization problem can be solved using the expectation-maximization family of algorithms by updating $\mathbf{\Sigma}^{-1}$ and $\mathbf{M}$ alternatively (Hallac et al., 2017).

**Some sensitivity analysis to hyperparameters.** The size of clusters isdetermined by a data-driven procedure following the original TICC work (i.e., it is determined with the highest silhouette score in clustering historical trajectories) (Hallac et al., 2017). To further address this concern, we ran an additional sensitivity analysis on the education domain by varying the number of TICC clusters around our default setting. In the main text we use K=14 clusters; we now compare against K=12 and 16, keeping all other TICC hyperparameters fixed to their default values (window size=1, switching sparsity=11e-2 (Hallac et al., 2017)) in all experiments to highlight that we use TICC as an off-the-shelf method rather than heavily tuned. We report the mean (std) student outcome: HBO 0.075 (.003), HBO (K=12) 0.071 (.006), HBO (K=16) 0.072 (.006). We observe that all HBO variants substantially outperform random and the best OPE/OPS baseline, while HBO (K=12) and HBO (K=16) are slightly better than FPS. This suggests that HBO is robust to reasonable changes in the number of clusters and that our conclusions do not hinge on a finely tuned clustering configuration. Such a finding is important and can be potentially generalized to common human-centric environments, and we plan to further pursue such an avenue in broader contexts both empirically and theoretically in the future. Importantly, TICC's clusters correspond to meaningful temporal motifs in human behavior (e.g. education and healthcare) noted by prior work (Hallac et al., 2017; Yang et al., 2025; Gao et al., 2024a), which align with our domains. As HBO is easy to be incorporated with any clustering algorithms, exploring advanced discretizations or continuous representations is an interesting direction for future work.

### E.2 Estimating Rewards and Next Observations

To calculate look ahead advantage at step $h$, we need to estimate rewards $\hat{r}_h$ received at current step $h$ by remaining on $\pi_{t<h}$, and next observations $\hat{o}_{h+1}$ by switching to policy $\pi^*_{t \geq h+1}$ at step $h+1$. In general, both parts can be estimated via model-based approaches (Hafner et al., 2020; Gao et al., 2022).

Specifically, in experiments for intelligent tutoring, we estimated both parts from the VAEs learned using historical trajectories under each policy, as in (Gao et al., 2022). To get $\hat{r}_h$, we selected the generated trajectory learned from the initial policy whose observation on step $h$ was closest to the current observation $o_h$ and used its reward on step $h$ as $\hat{r}_h$. The distance was calculated by Euclidean distance. We took averaged value if there were tie cases. To get $\hat{o}_{h+1}$, since the number of generated trajectories can be unlimited, we randomly rolled out 100 trajectories from the VAE learned using the target policy, and selected generated trajectories whose observations at step $h$ is closest to $o_h$, and used their next observations as $\hat{o}_{h+1}$. In experiments for sepsis treatment, we used the transition and reward matrices provided by (Namkoong et al., 2020), where they were learned to train the target policies that were used in our experiment.

## F Detailed Experimental Setup

### F.1 Training Resources

All experimental workloads are distributed among 9 1080ti(12G) and 10 a4000(16G) graphics cards.

### F.2 More Experimental Design

The student simulator was built from a dataset of 1,342 students, where each student received one of 3 intelligent tutoring systems decision policies, which guided the interactions with the student. Each student was randomly assigned one policy from the set of policies. The chi-squared test was

employed to check the relationship between policy assignment and subgroups, and it showed that the policy assignment cross subgroups were balanced with no significant relationship (p-value=0.479).

In intelligent tutoring historical data, each student only experienced one policy. To investigate the effects of policy switching, we built a simulator that models the dynamics of student-system interactions under corresponding policies following prior works (Gao et al., 2024a; 2022). At test time, we take the partial trajectory of a new student, and then, if the algorithm being evaluated suggests changing their policy, we then use our simulated dynamics model to roll out under the new policy. Specifically, we randomly rolled out 100 trajectories from the VAE learned using the target policy following the same distribution of initial states, and took selected generated trajectories whose observations at step h is closest to the current observation at step h (in our experiments we can find ones with Euclidean distance=0), and used their next observations as synthetic ones. Sepsis simulator has been widely adopted in existing works for policy learning and evaluation (Oberst & Sontag, 2019; Tang & Wiens, 2021; Namkoong et al., 2020).

**Oracle policies.** The oracle policy provided the optimal selection for each participant in hindsight by assuming at the beginning we could have picked for each participant the best policy as if the trajectory rollouts under all policies were available. Note that oracle is impossible to achieve as it depends on access to future observations, but it can inform approximately the best returns that can be achieved by HBO. Therefore the oracle policy should be viewed as an upper bound. We will make sure to clarify this in the updated text. In our experiments, the trajectory rollouts followed the same initial states as the test set used by HBO, and we operated the policy selected by HBO for each individual from their initial states.

### F.3 IMPLEMENTATIONS AND HYPER-PARAMETERS

#### F.3.1 INTERESTED SUB-GROUPS TO EXTRACT CBPs

To extract CBPs, in intelligent tutoring, we focused on the students whose final outcomes were less than or equal to average scores in training set. In sepsis treatments, we focused on patients who were not discharged from in-hospital stay until the end of horizon.

#### F.3.2 VARIATIONAL AUTO-ENCODER (VAE) FOR GENERATING SYNTHETIC DATA

To generate synthetic trajectories, for the components involving LSTMs, which include the encoder $q_\alpha(z_t|z_{t-1}, a_{t-1}, s_t)$, and $p_\eta(z_t|z_{t-1}, a_{t-1})$ in decoder, their architecture include one LSTM layer with 64 nodes, followed by a dense layer with 64 nodes. All other components do not have LSTM layers involved, so they are constituted by a neural network with 2 dense layers, with 128 and 64 nodes respectively. The output layers that determine the mean and diagonal covariance of diagonal Gaussian distributions use linear and softplus activations, respectively. The ones that determine the mean of Bernoulli distributions (*e.g.*, for capturing early termination of episodes) are configured to use sigmoid activations. For training, in sub-groups with sample size greater than 200 (*e.g.*, students assigned the initial policy), the maximum number of iteration is set to 1000 and minibatch size set to 64, and 200 and 4 respectively for subgroups with sample size less than or equal to 200 (*e.g.*, students assigned the RL-induced policies). Adam optimizer is used to perform gradient descent. To determine the learning rate, we perform grid search among $\{1e-4, 3e-3, 3e-4, 5e-4, 7e-4\}$. Exponential decay is applied to the learning rate, which decays the learning rate by 0.997 every iteration.

## G MORE RELATED WORKS

**RL for HCSs.** In modern HCSs, RL has raised significant attention toward enhancing the experience of human participants. Previous studies have demonstrated that RL can induce interactive education policies (Abdelshiheed et al., 2023; Mandel et al., 2014; Shen & Chi, 2016; Wang et al., 2017). For example, Zhou et al. (Zhou et al., 2022) applied hierarchical reinforcement learning (HRL) to improve students' normalized learning gain in a Discrete Mathematics course, and the HRL-induced policy was more effective than the Deep Q-Network induced policy. Similarly, in healthcare, RL has been used to synthesize policies that can adapt high-level treatment plans (Raghu et al., 2017; Namkoong et al., 2020; Lorberbom et al., 2021; Mandyam et al., 2023), or to control medical devices and surgical robotics from a more granular level (Gao et al., 2020; Lu et al., 2019; Richter et al., 2019).

Since online evaluation/testing is risky in practical HCSs, effective OPS methods are important in closing the loop, by significantly reducing the resources needed for online testing/deployment and preemptively justifying the safety of the policies subject to be deployed.

