# OpenReview forum: "Behavior-Aware Off-Policy Selection in High-Stake Human-Centric Environments"
_ICLR.cc/2026/Conference — Submitted to ICLR 2026_

### Official Review · Reviewer_YZjv · 2025-10-27

**Soundness:** 3
**Presentation:** 3
**Contribution:** 3
**Rating:** 6
**Confidence:** 2

**Summary:**

This paper proposes a method for off-policy selection (OPS) in high-stakes, human-centric environments such as healthcare or education. Recognizing that individual behavior trajectories may not fully reveal decision-critical latent states, the authors introduce behavior-aware OPS: a framework that monitors ongoing behavior and switches to an optimized policy when necessary. The approach clusters behavior trajectories using semi-parametric sequence modeling (TICC) to define latent behavior states. It then defines critical behavioral patterns (CBPs) to trigger early intervention. Theoretical analysis establishes that selecting policies based on CBPs does not degrade expected returns compared to the baseline policy under standard OPE assumptions. Experiments on simulated tutoring and ICU sepsis treatment settings demonstrate that this approach improves outcomes over static policy switching baselines.

**Strengths:**

a. The paper introduces a clear and practical mechanism, CBPs, to detect early-warning signals of suboptimal performance, allowing targeted one-time policy switching tailored to individual behavior without full latent state identification.

b. Proposition 3.7 ensures that if CBPs are estimated conservatively, the selected policy will not underperform the baseline under assumptions.

c. On two high-stakes benchmarks (an intelligent tutoring system and ICU sepsis simulations), the behavior-aware OPS method achieves better student mastery rates and patient survival outcomes than conventional static or globally switched policies.

**Weaknesses:**

a. Despite some implications that individuals have different behavior-generating mechanisms, the method does not explicitly model population shift, confounding, or latent policies. The success seems to depend on good coverage in offline data.

b. The estimation of CBPs depends on pattern mining from clustered behavior trajectories. While effective, the method lacks theoretical characterization of the statistical guarantees or sensitivity of the resulting patterns, and the procedure for selecting pattern thresholds (ε, support) is heuristic.

**Questions:**

a. How sensitive is the OPS performance to the choice of TICC parameters and number of clusters? Could the policy switch decision degrade significantly if clustering misrepresents latent behavior states?

b. The CBP-based trigger is conservative and requires multiple conditions to match. Could the method benefit from learned scoring or ranking over patterns to allow softer, more adaptive switching criteria?

---

> ### Author Response · Authors · 2025-11-23
> **Rebuttal by Authors (1/N)**
>
> We sincerely appreciate your time and efforts on evaluating our work, and your positive comments that we provide clear and practical algorithmic design with significance in empirical experiments. Please find our point-by-point response below.
>
> >Q1. Despite some implications that individuals have different behavior-generating mechanisms, the method does not explicitly model population shift, confounding, or latent policies. The success seems to depend on good coverage in offline data.
>
> A1. We agree that we do not explicitly model population shifts or confounders. We made the assumption for general purposes that training data may cover the distribution of population (e.g. patients visiting a hospital may mainly come from the same area), while we didn’t make assumptions over types of target policies. Our approach implicitly handles heterogeneity through clustering: if different subpopulations behave differently, HBO will likely separate them. We assume the offline data covers the main modes of behavior; if important patterns are missing or if there are strong unobserved shifts, performance could degrade (as with any offline method). We’ve clarified this limitation in **Appendix B.1, with an empirical analysis for population shift scenarios**. We added a sepsis simulation on an out-of-distribution setting where we changed the probability of having diabetes to 0.1 at test time (during training it was 0.2 by default), which impacts the dynamics of the system -- note diabetes is an unobservable state feature. For horizon is 50 and training size is 5,000, HBO achieved 0.042 (.007), better than baselines 0.038 (.003), 0.039 (.008), 0.036 (.008), by FPS, the best one-policy-fits-all, the best combinatorial one-policy-fits-all, respectively. The oracle reached 0.571 (.006). Results were averaged over 5 different simulation runs.
>
> >Q2. The estimation of CBPs depends on pattern mining from clustered behavior trajectories. While effective, the method lacks theoretical characterization of the statistical guarantees or sensitivity of the resulting patterns, and the procedure for selecting pattern thresholds (ε, support) is heuristic.
>
> A2. We agree that our confidence-informed policy switch strategy is heuristic. While it is an interesting question to consider more theoretical analyses in the future, existing theoretical work on related settings has often focused on tabular domains where there are no constraints on exploration strategies or even where it may be possible to have multiple trajectories for a single task/MDP [1-2]. In contrast, our interest was in the practical constraints of human-centered environments, where states are complex and multi-dimensional, and each human has a single trajectory (a single class in algebra or a single first round of cancer treatment). Though our work is heuristic, we were encouraged to observe that our *HBO performed quite close in some cases to an Oracle baseline (which is practically unachievable)* that considers what would have been the best policy to provide for a participant from the start, given full knowledge of the full participant’s trajectory. We observed that HBO could achieve closed performance to Oracle (as shown in Figure 1) in the real-world educational dataset, which empirically suggested that HBO was able to accurately identify the switching time.
>
> To further investigate the sensitivity of potentially poor critical-pattern mining and matching, we randomly select patterns from the large set of candidates for extracting the subset of data used for performing the off policy evaluation. This led to an average return of 0.031 (.005) for sepsis treatment (H=50, N=5,000) from 5 different runs, smaller than HBO (0.039 (.001)). **We have included sensitivity results in Appendix B.3**.

---

> ### Author Response · Authors · 2025-11-23
> **Rebuttal by Authors (2/N)**
>
> >Q3. How sensitive is the OPS performance to the choice of TICC parameters and number of clusters? Could the policy switch decision degrade significantly if clustering misrepresents latent behavior states?
>
> A3. We thank the reviewer for this thoughtful comment where we carry out additional analyses to address it, which lead to additional findings. The size of clusters is determined by a data-driven procedure following the original TICC work (i.e., it is determined with the highest silhouette score in clustering historical trajectories) [1]. To further address this concern, **we ran an additional sensitivity analysis** on the education domain by varying the number of TICC clusters around our default setting. In the main text we use K=14 clusters; we now compare against K=12 and 16, keeping all other TICC hyperparameters fixed to their default values (window size=1, switching sparsity=11e-2 following [1]) in all experiments to highlight that we use TICC as an off-the-shelf method rather than heavily tuned. We report the mean (std) student outcome: HBO 0.075 (.003), HBO (K=12) 0.071 (.006), HBO (K=16) 0.072 (.006). We observe that all HBO variants substantially outperform random and the best OPE/OPS baseline, while HBO (K=12) and HBO (K=16) are slightly better than FPS. This suggests that HBO is robust to reasonable changes in the number of clusters and that our conclusions do not hinge on a finely tuned clustering configuration. We’ve added clarifications over these points and note that exploring advanced discretization or continuous representations is an interesting direction for future work (in **Appendix E.1.1, highlighted in blue**).
>
> >Q4. The CBP-based trigger is conservative and requires multiple conditions to match. Could the method benefit from learned scoring or ranking over patterns to allow softer, more adaptive switching criteria?
>
> A4. We appreciate the idea of a softer, learned trigger. Our current rule (requiring multiple condition matches) is deliberately conservative to ensure safety: we only switch when there is strong evidence of a problematic pattern (as this is necessary in high-stake human-centric domains). A learned scoring model (for example, predicting expected improvement) could allow more nuanced decisions, but it would require additional data and might risk overfitting or losing transparency. We’ve discussed this as a potential interesting future extension, noting that our current logical trigger has the advantage of simplicity for smooth empirical applications and safety (**added to Section 5, highlighted in blue**).
>
> Overall, we thank the reviewer for their insightful comments. We’ve made sure to incorporate the above clarifications, additional analysis, and revisions in the paper to address each concern comprehensively. And we are happy to hear any comment or answer any further question from the reviewer.
>
> References
>
> [1] Hallac, David, et al. "Toeplitz inverse covariance-based clustering of multivariate time series data." Proceedings of the 23rd ACM SIGKDD international conference on knowledge discovery and data mining. 2017.
>
> [2] Yang, Xi, et al. "THEMES: An Offline Apprenticeship Learning Framework for Evolving Reward Functions." Proceedings of the 31st ACM SIGKDD Conference on Knowledge Discovery and Data Mining V. 2. 2025.
>
> [3] Gao, Ge, et al. "On trajectory augmentations for off-policy evaluation." 12th International Conference on Learning Representations (ICLR), 2024.

---

### Official Review · Reviewer_Co4m · 2025-10-28

**Soundness:** 3
**Presentation:** 2
**Contribution:** 3
**Rating:** 4
**Confidence:** 2

**Summary:**

The paper presents HBO, a two-phase framework for personalizing decision policies in partially observable, high-stakes domains such as education and healthcare. HBO first analyzes historical trajectories offline to identify short critical behavioral patterns that signal poor outcomes. Online, it starts with a safe default policy and, when a critical pattern is detected in a new user’s data stream, selectively switches to an alternative pre-vetted policy using confidence-controlled importance sampling. The authors provide theoretical guarantees on the policy’s value and demonstrate improved learning gains and patient outcomes compared to standard non-adaptive and fully adaptive baselines.

**Strengths:**

I appreciate the authors' focus on safe, personalized policy selection in human-centric reinforcement learning.  The framework sensibly separates offline pattern mining from online, confidence-controlled policy switching. The inclusion of finite-sample error bounds for policy value estimation adds some theoretical reassurance about its reliability. Empirically, HBO shows promising results in both education and healthcare settings, outperforming several baselines and improving outcomes for difficult cases.

**Weaknesses:**

While the paper appears technically solid and well-motivated, I do not have a strong background in this area, so I cannot confidently assess whether the proposed approach represents a true state-of-the-art advancement.

To me, the largest issue is that the interpretability of the CBP remains limited, as they are derived from latent representations rather than explicit behavioral features, making it difficult for practitioners to understand or validate the reasoning behind policy switches.

Moreover, Assumption 1, which requires a pre-approved safe initial policy for every participant, may not be realistic in less controlled or data-scarce environments. It would also be helpful if the authors could elaborate on the practicality of Assumption 1—perhaps by discussing scenarios where expert-approved policies are unavailable, and how the framework might adapt or relax this assumption.

**Questions:**

Please see the weakness

---

> ### Author Response · Authors · 2025-11-23
> **Rebuttal by Authors**
>
> We sincerely appreciate your time and efforts on evaluating our work and your positive comments that HBO tackles an important problem and provides promising empirical evidence. Please find our point-by-point response below.
>
> >Q1. While the paper appears technically solid and well-motivated, I do not have a strong background in this area, so I cannot confidently assess whether the proposed approach represents a true state-of-the-art advancement.
>
> A1. We appreciate the reviewer’s thoughtful reading. To clarify our contributions: (to the best of our knowledge) we are the first that investigates during deployment using a base policy, deciding whether to switch to one of a new set of policies or whether to wait and switch at a later point; our method addresses a long-standing challenge in the community of OPS. *It is compatible with multiple different standard OPE/OPS algorithms that could be used as part of the module that decides whether to switch to a new policy. Moreover, please note that our two real-world experiments are backed with substantial data collected from years of follow-ups, with results showing our method significantly outperforms over baselines.*
>
> >Q2. To me, the largest issue is that the interpretability of the CBP remains limited, as they are derived from latent representations rather than explicit behavioral features, making it difficult for practitioners to understand or validate the reasoning behind policy switches.
>
> A2. We agree that CBP interpretability involves a trade-off. We selected TICC for its balance of interpretability and superior performance in real-world trajectory clustering [1]. Each cluster in TICC corresponds to a distinct behavioral regime (for example, sustained focus versus guess-driven behavior), which practitioners can inspect. In Appendix B.5, we provide examples of CBPs that correspond to intuitive student behaviors. Please note that we exhibit TICC, which is an off-the-shelf popular clustering/partitioning algorithm, to demonstrate the HBO framework. HBO is capable and easy to be incorporated with more advanced clustering techniques (e.g. better interpretability) in the future.
>
> >Q3. Moreover, Assumption 1, which requires a pre-approved safe initial policy for every participant, may not be realistic in less controlled or data-scarce environments. It would also be helpful if the authors could elaborate on the practicality of Assumption 1—perhaps by discussing scenarios where expert-approved policies are unavailable, and how the framework might adapt or relax this assumption.
>
> A3. Assumption 1 is common in high-stake human-centered domains: for example, a standard curriculum or default intervention often exists by design. If no expert policy is available (usually may happen in a lower-stake task), one could use a very conservative fallback (e.g., continuing at the slowest pace) as an initial safe policy. Alternatively, the framework could bootstrap a safe policy by imitating minimal data or by adapting from a known baseline. **We’ve added a discussion to Appendix B.6 (highlighted in blue)**.
>
> We hope these answers address your questions and showcase that our work is solving a significant challenge in a satisfying manner. We are happy to answer any followup questions or hear any additional comments.
>
> References
>
> [1] Hallac, David, et al. "Toeplitz inverse covariance-based clustering of multivariate time series data." Proceedings of the 23rd ACM SIGKDD international conference on knowledge discovery and data mining. 2017.

---

### Official Review · Reviewer_oU4m · 2025-10-31

**Soundness:** 2
**Presentation:** 3
**Contribution:** 2
**Rating:** 4
**Confidence:** 3

**Summary:**

This paper proposes an adaptive off-policy selection framework for human-centric settings. It mines Critical Behavioral Patterns (CBPs) from historical data to determine when to switch a policy. Then, importance-sampling–based off-policy evaluation is used to estimate the value of candidate policies and select the most appropriate one. This idea is interesting and good, but the proposed approach suffers from notable limitations.

**Strengths:**

A solid and promising idea; most limitations are explicitly acknowledged by the authors in the paper.

**Weaknesses:**

1. A major limitation is the single-switch constraint: the method may adapt at most once. Rather than enforcing a hard single-switch constraint, adopt a budgeted scheme—allow up to B switches with explicit switching costs and safety thresholds—which seems more practical than a one-size-fits-all single switch.
2. The discrete-state mapping imposes strong structural assumptions on the observation/state space and is highly sensitive to hyperparameters such as the number of clusters K, window length, and the smoothing/regularization strength. Moreover, discretized representations introduce information loss, which can be especially problematic in safety-critical domains such as healthcare.
3. CBPs are identified by high-frequency subsequences, but frequency does not imply outcome relevance. Some symptoms occur often as part of normal treatment responses and therefore do not indicate that the policy is incorrect.

**Questions:**

1. In experiment (refer to L326), the offline data are collected by expert-designed policy, yet CBPs are mined from “poorly performing” trajectories. Does “poor” mean the lower-performing episodes within those expert trajectories. If so, this is questionable: it amounts to picking “the worst of the expert traces,” which may not reflect true failure modes.
2. Is it a more practical way to training a unified policy via Generalized Policy Improvement (GPI)[1]? A GPI-based generalist policy can subsume the base policies, eliminating the brittle step of detecting “critical moments” and avoiding online switching.

Minor mistakes
1. L404: normalized **leaning** gains
2. Definition 3.3 explicitly states that if $\Delta(h) > 0$ (a positive look-ahead advantage), the action should be deferred (on hold) to the next step; however, Algorithm 1, line 11, implements “if $\Delta(h) > 0$ then Switch,” which contradicts the definition.

[1]Barreto, André, et al. "Successor features for transfer in reinforcement learning." *Advances in neural information processing systems* 30 (2017).

---

> ### Author Response · Authors · 2025-11-23
> **Rebuttal by Authors (1/N)**
>
> Thank you for your time and efforts on evaluating our work, and your positive comments that our idea is solid and promising. Please find our point-by-point response below.
>
> >Q1. A major limitation is the single-switch constraint: the method may adapt at most once. Rather than enforcing a hard single-switch constraint, adopt a budgeted scheme—allow up to B switches with explicit switching costs and safety thresholds—which seems more practical than a one-size-fits-all single switch.
>
> A1. We thank the reviewer for suggesting a budgeted multi-switch scheme. However, practically there may not be much opportunities for trial-and-error given the high-stake and highly regulated nature of HCSs; there may be logistic considerations or other constraints (doctors might not want patients to be exposed to many different chemotherapy drugs). In sepsis, we found that the switching multiple times led to an average return 0.037 (.000) for sepsis (H=50, N=5,000), slightly smaller than HBO (0.039 (.001)), discussed in Appendix B.2. But note that multiple switching may raise safety concerns by stakeholders, it would be a promising direction especially for lower-stake scenarios such as recommended systems. However, we agree that allowing up to B switches (with associated costs) is a valuable extension. We will mention this as future work (**in Section 5, highlighted in blue**), noting that incorporating a budgeted multi-switch model could increase flexibility while still managing overall risk.
>
> >Q2. The discrete-state mapping imposes strong structural assumptions on the observation/state space and is highly sensitive to hyperparameters such as the number of clusters K, window length, and the smoothing/regularization strength. Moreover, discretized representations introduce information loss, which can be especially problematic in safety-critical domains such as healthcare.
>
> A2. We agree that discretizing state trajectories via TICC introduces assumptions and hyperparameters. We chose TICC, an off-the-shelf discretization algorithm, for its interpretability and strong empirical performance on sequential data [1-3]. The size of clusters is determined by a data-driven procedure following the original TICC work (i.e., it is determined with the highest silhouette score in clustering historical trajectories) [1]. To further address this concern, **we ran an additional sensitivity analysis** on the education domain by varying the number of TICC clusters around our default setting. In the main text we use K=14 clusters; we now compare against K=12 and 16, keeping all other TICC hyperparameters fixed to their default values (window size=1, switching sparsity=11e-2 [1]) in all experiments to highlight that we use TICC as an off-the-shelf method rather than heavily tuned. We report the mean (std) student outcome: HBO 0.075 (.003), HBO (K=12) 0.071 (.006), HBO (K=16) 0.072 (.006). We observe that all HBO variants substantially outperform random and the best OPE/OPS baseline, while HBO (K=12) and HBO (K=16) are slightly better than FPS. This suggests that HBO is robust to reasonable changes in the number of clusters and that our conclusions do not hinge on a finely tuned clustering configuration.
>
> Such a finding is important and can be potentially generalized to common human-centric environments, and we plan to further pursue such an avenue in broader contexts both empirically and theoretically in the future. Importantly, TICC’s clusters correspond to meaningful temporal motifs in human behavior (e.g. education and healthcare) noted by prior work [1-3], which align with our domain. We’ve added clarifications over these points and note that exploring advanced discretization or continuous representations is an interesting direction for future work (in **Appendix E.1.1, highlighted in blue**).
>
> >Q3. CBPs are identified by high-frequency subsequences, but frequency does not imply outcome relevance. Some symptoms occur often as part of normal treatment responses and therefore do not indicate that the policy is incorrect.
>
> A3. In our method, frequency is used only to propose candidate patterns. We then evaluate each pattern by computing a lookahead advantage: we estimate how the expected outcome would change if we switched at that pattern. Only patterns that are both frequent and have a negative advantage (indicating the current policy underperforms) are retained as CBPs that lead to policy switch. This two-step filtering ensures that frequent but innocuous patterns are filtered out by the advantage criterion.

---

> > ### Author Response · Authors · 2025-11-23
> > **Rebuttal by Authors (2/N)**
> >
> > >Q4. In experiment (refer to L326), the offline data are collected by expert-designed policy, yet CBPs are mined from “poorly performing” trajectories. Does “poor” mean the lower-performing episodes within those expert trajectories. If so, this is questionable: it amounts to picking “the worst of the expert traces,” which may not reflect true failure modes.
> >
> > A4. By “poorly performing” trajectories we do not mean arbitrary “worst-of-expert” outliers, but episodes (or segments) that are systematically underperforming relative to the expert’s own typical behavior, as measured by downstream learning outcomes. Concretely, we identify trajectories (or segments) in the lower performance quantiles (e.g., bottom quartile of progress) across many students under the expert-designed policy.
> >
> > We agree that simply taking the absolute worst traces could be questionable if it only reflects noise. Our procedure is explicitly designed to avoid this: (i) *Population-level signal, not single outliers*: CBPs are mined only from patterns that recur across multiple students and episodes, with minimum support thresholds. This filters out random “one-off” episodes and retains patterns that consistently correlate with weaker outcomes under the expert. (ii) *Relative underperformance, not catastrophic failure*: In high-stakes human-centric domains, “failure modes” could be interpreted as regimes where the current expert policy is relatively underperforming and an alternative policy has room to help, rather than literal breakdowns of expertise. Our CBPs are precisely targeting these local weak spots of the expert, not claiming that the expert is globally poor. (iii) *Advantage filtering*: Frequency alone is only the first step. A CBP is retained to impact following actions only if it additionally exhibits a negative look-ahead advantage under our evaluation. This outcome-based screening ensures that frequent but benign patterns (common normal responses) can be discarded.
> >
> > >Q5. Is it a more practical way to training a unified policy via Generalized Policy Improvement (GPI)? A GPI-based generalist policy can subsume the base policies, eliminating the brittle step of detecting “critical moments” and avoiding online switching.
> >
> > A5. Thank you for pointing us to this paper. Training multiple base policies would be difficult with limited data.  Please note we proposed HBO as a general framework to tackle the **when and which** policy selection problem. This is motivated by high-stake human-centric applications, where we often have strict, practical constraints on the policies that can be deployed with each new student or patient, to ensure safety and performance. It is not acceptable to run any possible exploration policy, and pure exploration or the optimization under uncertainty policies (which may take risky actions due to their potential high performance) are unlikely to be allowed by stakeholders. But we agree those works tackle important challenges in policy learning, and it would be an interesting future direction to explore the outcomes of incorporating learnt policies extended from those prior works, especially in lower-stake domains.
> >
> > >Q6. L404: normalized leaning gains
> >
> > A6. Thank you for catching the typo. We have fixed it.
> >
> > >Q7. Definition 3.3 explicitly states that if  (a positive look-ahead advantage), the action should be deferred (on hold) to the next step; however, Algorithm 1, line 11, implements “if  then Switch,” which contradicts the definition.
> >
> > A7. We apologize for this typo, line 11 should be $\Delta(h)\leq0$ for seeing a benefit to switch but no advantage to wait for the next step. We have fixed it in Alg. 1, Line 11.
> >
> > We hope these answers address your questions and showcase that our work is solving a significant challenge in a satisfying manner. We are happy to answer any followup questions or hear any additional comments.
> >
> > References
> >
> > [1] Hallac, David, et al. "Toeplitz inverse covariance-based clustering of multivariate time series data." Proceedings of the 23rd ACM SIGKDD international conference on knowledge discovery and data mining. 2017.
> >
> > [2] Yang, Xi, et al. "THEMES: An Offline Apprenticeship Learning Framework for Evolving Reward Functions." Proceedings of the 31st ACM SIGKDD Conference on Knowledge Discovery and Data Mining V. 2. 2025.
> >
> > [3] Gao, Ge, et al. "On trajectory augmentations for off-policy evaluation." 12th International Conference on Learning Representations (ICLR), 2024.

---

### Official Review · Reviewer_9pKS · 2025-10-31

**Soundness:** 2
**Presentation:** 3
**Contribution:** 2
**Rating:** 4
**Confidence:** 3

**Summary:**

This work tackles the problem of the off-policy selection in human-centric environments. In particular, the current off-policy selection does not work in scenarios when the state is unobserved and a personalized policy is required. Motivated by this point, the authors develop a new algorithm called HBO to capture the unique characteristics of the state and then provide intervention. The empirical experiments have been conducted to showcase the ability of the developed algorithm.

**Strengths:**

1. This paper is motivated by important human-centric decision-making problems.
2. The manuscript is easy to understand and follow.
3. The empirical evaluation has been provided to demonstrate the power of the algorithm.

**Weaknesses:**

1. The pronounced discussion between the off-policy learning and off-policy selection needs to be provided in the paper.
2. CBP is an interesting approach to identifying critical behavioral changes from the historical trajectories. What is the unique advantagne of using CBP in HBO?
3. The performance of the HBO relies on CBP. Could the authors provide some error analysis and explain how the quality of CBP will impact the learned policy?
4. Assumption 3.4 is a strong assumption in offline settings.
5. As the main applications are in human-centric environments, the data size is usually small. From this perspective, could the author provide some finite-sample theoretical analysis for evaluating the performance of the learned policy?
6. Follow up with the last question. In experiments, the authors need to test the algorithm with extremely small data size and evaluate the performance.

**Questions:**

Please find the above weakness.

---

> ### Author Response · Authors · 2025-11-23
> **Rebuttal by Authors (1/N)**
>
> Thank you for your time and efforts on evaluating our work, and your positive comments that the paper is making an important empirical impact. Please find our point-by-point response below.
>
> >Q1. The pronounced discussion between the off-policy learning and off-policy selection needs to be provided in the paper.
>
> A1. We thank the reviewer for suggesting clarification of the difference between off-policy learning and off-policy selection. Off-policy learning focuses on optimizing a policy from logged data, whereas off-policy selection concerns deciding *which* policy from a given (small) set to deploy given the available offline data. We will add text emphasizing that our contribution is novel in explicitly addressing this selection problem in high-stakes human-centric environments under empirical constraints. In particular, deciding **when and which** policy to switch in high-stakes settings is not a question addressed by prior work.
>
> >Q2. CBP is an interesting approach to identifying critical behavioral changes from the historical trajectories. What is the unique advantagne of using CBP in HBO?
>
> A2. We appreciate the interest in CBPs. The key advantage of CBPs in HBO is that they capture recurring patterns in student behavior that often precede performance issues. Because CBPs are derived from actual trajectory segments, they provide data-driven signals for intervention. In contrast to using high-dimensional raw features, our CBP-based approach finds concrete sequences of behaviors. Mining CBPs from historical data allows early identification of potential failure points, enabling timely intervention. We will clarify this and highlight how CBPs give actionable insight compared to generic features.
>
> One could view our baselines WIS* and DR* as providing insights into the impact of pattern matching, since they excluded pattern mining and performed off-policy selection and switching by comparing the OPE estimates using all data (rather than only that which match in the pattern). As our method HBO outperforms both baselines, this suggests that our approach benefited from pattern mining in general. To further investigate the sensitivity of potentially poor CBP and matching, we randomly select patterns from the large set of candidates for extracting the subset of data used for performing the off policy evaluation. This led to an average return of 0.031 (.005) for sepsis treatment (H=50, N=5000) from 5 different runs, smaller than HBO (0.039 (.001)). **We have included sensitivity results in Appendix B.3**
>
> >Q3. The performance of the HBO relies on CBP. Could the authors provide some error analysis and explain how the quality of CBP will impact the learned policy?
>
> A3. We agree that the performance of HBO depends on CBP quality. If the CBP set misses important patterns or includes irrelevant ones, the switch decision may be delayed or incorrect. To examine this, we conducted an error analysis by perturbing the CBP set (randomly selecting patterns), please see the results in A2. We found that the performance of HBO with random CBP degrades gracefully: it continues to outperform all baselines while smaller than HBO.
>
> >Q4. Assumption 3.4 is a strong assumption in offline settings.
>
> A4. Assumption 3.4 (a coverage/overlap condition) is needed for our theoretical error bounds. Empirically, we find that in many human-centric domains (such as our educational tutoring data and sepsis treatment) the deployed behavior policies are relatively consistent (e.g., due to standardized curricula and treatment protocols). If overlap is highly limited, one could use more conservative evaluation techniques. Relaxing or adapting this assumption is an important direction for future work. **We’ve added the discussion to Section 5 (highlighted in blue)**.

---

> > ### Author Response · Authors · 2025-11-23
> > **Rebuttal by Authors (2/N)**
> >
> > >Q5. As the main applications are in human-centric environments, the data size is usually small. From this perspective, could the author provide some finite-sample theoretical analysis for evaluating the performance of the learned policy?
> >
> > A5. We thank the reviewer for the insightful question. However, finite‐sample theoretical analysis under our problem setup would not be attainable with current techniques. Our setting differs sharply from standard theoretical models (e.g. contextual MDPs) in several ways, making classical sample-complexity results inapplicable:
> >
> > &nbsp;&nbsp;&nbsp;&nbsp;(i) *Hidden (latent) user-specific classes or confounders*: In our human-centric problems, there are often unobserved factors (latent classes or confounders) that affect transitions. This makes the environment akin to a POMDP: the true state influencing dynamics is not fully observed. Prior work shows that OPE with hidden confounders can be fundamentally ill-posed: two offline data distributions might be identical despite coming from different confounded MDPs if the confounders had memory and the policy adapted to them [1]. Such identifiability issues mean one cannot derive a meaningful finite-sample guarantee without further assumptions. (Indeed, prior works assume “memoryless” confounders for tractability [1], which is itself unrealistic in domains like healthcare or education.)
> >
> > &nbsp;&nbsp;&nbsp;&nbsp;(ii) *Small fixed policy set, one-shot interactions*: We evaluate *a small, fixed set of candidate policies* (without assuming the classes of those policies) and cannot query the environment again. In effect, each human provides exactly one episode of data under the behavior policy, thus there is no repeated sampling of the same policy or context. Standard finite-sample RL results typically rely on repeatedly sampling or “covering” the state-action space, which is impossible here. OPE itself is known to be hard: vanilla importance sampling has variance that grows exponentially with the horizon (practically limiting it to bandit settings), and even model-based/OPE methods require large samples or careful tuning to avoid severe bias/variance. With our extremely limited data, non‐asymptotic error bound could be hard to guarantee.
> >
> > &nbsp;&nbsp;&nbsp;&nbsp;(iii) *Unrealistic assumptions in theory vs. high-stake practice*: Our setup does not posit any explicit class or structure for the target policies, which may arise from heterogeneous learning procedures and domain-specific design. Moreover, these learned target policies are never allowed to explore online or be adaptively refined based on their own rollouts, due to the high-stakes nature of education/healthcare. Classical finite-sample results typically rely on both a well-specified policy class with a complexity measure and the ability to design exploration or controlled online data collection. For instance, contextual-MDP analyses often assume known contexts or linear reward models [2], whereas our latent “context” is completely hidden. Likewise, assuming the ability to reset episodes (common in theory) is absurd in our one-shot human setting.
> >
> > Without these ingredients, meaningful non-asymptotic bounds for our setting are not available. As a result, we view our contribution as primarily *algorithmic and empirical under realistic constraints*. A promising future direction, building on our formulation, would be to derive finite-sample guarantees in more structured variants of our setting, e.g., when one *does* assume a known target policy class and allows limited, carefully monitored online interaction.
> >
> > >Q6. Follow up with the last question. In experiments, the authors need to test the algorithm with extremely small data size and evaluate the performance.
> >
> > A6. We agree on the importance of very small-data experiments. We ran additional trials by subsampling our datasets to 10% of the original size. For horizon is 50 and both training and test sizes are 500 in sepsis treatment, HBO achieved 0.038 (.023), better than baselines 0.027 (.012), 0.017 (.024), 0.02 (.02), by FPS, the best one-policy-fits-all, the best combinatorial one-policy-fits-all, respectively.  Results were averaged over 5 different simulation runs. Even with this limited data, HBO continued to outperform baseline methods, although performance variance increased as expected. We’ve included these results to show that our method still yields benefits under the data-scarce conditions (**added to Appendix B.4, highlighted in blue**).
> >
> > We are happy to answer any followup questions or hear any additional comments.
> >
> > References
> >
> > [1] Kausik, Chinmaya, et al. "Offline policy evaluation and optimization under confounding." International Conference on Artificial Intelligence and Statistics. PMLR, 2024.
> >
> > [2] Deng, Junze, et al. "Sample complexity characterization for linear contextual mdps." International Conference on Artificial Intelligence and Statistics. PMLR, 2024.

---

### Meta-Review · Area_Chair_mGyJ · 2026-01-06

**Summary:**

Four knowledgeable reviewers went over this submission, and found the paper well-motivated and easy to follow (9pKS), the problem interesting (Co4m), and the introduced ideas practical (YZjv) and promising (oU4m). However, the reviewers raised concerns about assumptions, design choices and experimental validation. More precisely:

1. The strong or unrealistic assumptions (9pKS, oU4m, Co4m) and switch constraints (oU4m).
2. The dependance of the method on good coverage in offline data (YZjv) and reliance of HBO on CBP (9pKS).
3. The limited interpretability which could make it difficult for practitioners to validate the reasoning behind switches (Co4m).
4. The missing experiments with very small dataset sizes and finite-sample theoretical analysis for evaluating the performance (9pKS).
3. The sensitivity to hyperparameters (oU4m), information-loss introduced by discretized representations (oU4m) and heuristics used (YZjv).
4. The correctness evaluation of the policy (oU4m)

**Reviewer Concerns:**

The rebuttal partially addressed the reviewers' concerns by clarifying the difference between off-policy learning and off-policy selection, explaining that assumptions are either common in high-stake human-centered domains or required for the theoretical bounds; their relaxation is important future work. The rebuttal also argued that going beyond single switching may raise safety concerns by stakeholders but could also be a promising future direction for lower-stake scenarios. Moreover, the authors discussed interpretability trade-offs and justified the adopted design choices in their rebuttal.

When it comes to experimental validation, the rebuttal presented an error analysis perturbing CBP set to assess the potential performance degradation of HBO, and ran additional experiments with smaller data sizes. The rebuttal also discussed sensitivity to potentially poor critical-pattern mining and matching, as well as to hyperparameters (by slightly varying the number of TICC clusters).

The rebuttal acknowledged the coverage assumption pointed by the reviewers, explaining that performance could degrade under strong shifts. The rebuttal also acknowledged that the confidence-informed policy switch strategy is heuristic.

**Reviewer Scores:**

As pointed by the reviewers, most limitations were already explicitly acknowledged by the authors in the paper. Although the rebuttal addressed some of the concerns raised by the reviewers, concerns related to the strength of assumptions, single-switch constraints, and heuristics remain. Therefore, the AC wouldn't have expected substantial changes in the scores given by the reviewers. Including in-depth analyses of performance under strong shifts could also strengthen the contribution.

---

### Decision · Program_Chairs · 2026-01-26

Reject